# Minimalist Softmax Attention Provably Learns Constrained Boolean Functions

## Abstract

We study the computational limits of learning $k$-bit Boolean functions (specifically, AND, OR, and their noisy variants), using a minimalist single-head softmax-attention mechanism, where $k = \Theta(d)$ relevant bits are selected from $d$ inputs. We show that these simple AND and OR functions are unsolvable with a single-head softmax-attention mechanism alone. However, with *teacher forcing*, the same minimalist attention is capable of solving them. These findings offer two key insights: Architecturally, solving these Boolean tasks requires only *minimalist attention*, without deep Transformer blocks or FFNs. Methodologically, one gradient descent update with supervision suffices and replaces the multi-step Chain-of-Thought (CoT) reasoning scheme of [Kim and Suzuki, ICLR 2025] for solving Boolean problems. Together, the bounds expose a fundamental gap between what this minimal architecture achieves under ideal supervision and what is provably impossible under standard training.

## 1 Introduction

We study the computational limits of learning monotone $k$-bit Boolean functions (i.e, AND/OR with $k$ relevant bits) with $d$-bit input using a minimalist one-head softmax-attention layer. In particular, we show that a *single softmax-attention head* provably learns an unknown $k$-bit AND/OR function, where $k = \Theta(d)$, after *one gradient step* if the training loss includes a teacher-forcing signal. In contrast, under ordinary end-to-end training (only input-label pairs, no intermediate hints) *no* algorithm running in $\mathrm{poly}(d)$ time can recover the same function, even when given $e^{\Omega(d)}$ examples.

Transformers dominate modern machine learning (Devlin et al., 2018; Brown et al., 2020; Floridi & Chiriatti, 2020; Ji et al., 2021; Touvron et al., 2023a;b; Zhou et al., 2023; 2024; 2025), yet their precise capabilities and limits remain elusive. For instance, Large Language Models can achieve human-level reasoning ability in expert problems (Singhal et al., 2023; Bubeck et al., 2024; Gao et al., 2025), but fail simple arithmetic problems (Li et al., 2024b; Chiang, 2024; Mahendra et al., 2025). Similarly, Transformer-based generative models, such as Diffusion Transformers (DiTs) (Peebles & Xie, 2023), can generate high-quality realistic visual content (Saharia et al., 2022; Ho et al., 2022; Wu et al., 2023), but they may fail at simple counting tasks or basic physical constraints (Huang et al., 2023a; Guo et al., 2025a;b). Thus, studying what tasks a Transformer can or cannot learn is both theoretically intriguing and practically important. On one hand, identifying inherent weaknesses can guide the design of more robust architectures and training methods (e.g., (Hu et al., 2025) identify necessary conditions for fast LoRA). On the other hand, uncovering new capabilities of even simplified Transformer components can expand our understanding of their potential (e.g., (Kajitsuka & Sato, 2023; 2024) establish universality of simple transformers and transformers' minimal requirements for memorizing a set of data). Many theoretical works chart this landscape, yet Transformers' training dynamics on algorithmic or logical problems remain underexplored.

Recently, (Kim & Suzuki, 2025) show that a one-layer Transformer can solve the parity function efficiently *when* provided with intermediate Chain-of-Thought (CoT) reasoning steps (i.e., with *teacher forcing*), but struggles to learn parity via end-to-end training without such assistance. These findings highlight a *supervision-gap* question: the choice of training regime alone can lead to distinct learning behavior in the same model. This contrast motivates a deeper investigation into the conditions under which Transformer-like architectures succeed or fail on structured tasks.

In this work, we investigate whether the same supervision gap already appears for the simpler $k$-Boolean problem (i.e., AND/OR) and whether a even simpler architecture (one single-head attention *without* FFN) can still close it. This simple "$k$-Boolean" task serves as a proxy for understanding how gradient-based training can (or cannot) discover important features and compute logical operations in a minimalist attention network. Formally, the target function is an unknown $k$-bit AND/OR with $k = \Theta(d)$ over $d$ binary inputs. The model is nothing more than a single-head softmax-attention layer — no feed-forward layer — starting with no clue which $k$ positions matter. Then, we ask:

> Can gradient descent training on input-output examples learn to attend to the correct $k$ bits and reliably compute the AND/OR?

Our analysis yields both a provably efficient learning result and a hardness result.

**Theorem 1.1** (Upper bound (Efficient Learnability with Teacher Forcing), Informal Version of Theorem 4.1)**.** *With intermediate supervision that exposes the Boolean label during training, the initial gradient already aligns with the indicator of the true feature subset. A single gradient update is enough to drive the model's attention weights to the correct $k$ positions, yielding vanishing classification error.*

**Theorem 1.2** (Lower bound (Intractability under End-to-End Training), Informal Version of Theorem 4.3)**.** *Remove that hint and the picture flips: the gradient of the usual loss averages over $\binom{d}{k}$ competing hypotheses and is therefore nearly uninformative. We prove that any learner, regardless of step size, adaptivity, or loss landscape access, fails to identify the relevant bits even after $e^{\Theta(d)}$ samples.*

**Contributions.** These results reveal a dramatic gap between what is achievable with the right supervision and what is provably impossible with naive training. Our contributions are two-fold:

- **Upper bound (Theorem 1.1).** We prove that if the model is trained with intermediate supervision (a form of teacher forcing where the model is guided to correctly compute partial results), then just *one step* of gradient descent from a random initialization suffices to identify the correct $k$-bit subset and achieve low error. In fact, with $n = \Omega(d^{\varepsilon})$ samples for any constant $\varepsilon > 0$, a single gradient update can drive the classification error to $O(d^{-\varepsilon/8})$. This result shows that, under the right training regime, even a one-layer attention mechanism can rapidly learn a high-dimensional conjunction or disjunction. In other words, *one-layer attention is in principle powerful enough to implement the required logical function, and it can do so with minimal training when given appropriate hints.*

- **Lower bound (Theorem 1.2).** In contrast, we prove a strong lower bound for the standard end-to-end training setting with no intermediate signals. Intuitively, without chain-of-thought style guidance, the learning algorithm must discover the relevant $k$ bits and the correct logical operation purely from input-output examples, which poses a computationally hard search problem. We show that any algorithm (in particular, any gradient-based learner) will *fail* to recover the correct subset of bits, even if it is given as many as $n = \exp(\Theta(d))$ training examples. Equivalently, with standard training the model's error remains bounded away from zero unless it executes a super-polynomial (exhaustive) search. This lower bound relies on constructing challenging initializations/loss landscapes that effectively trap polynomial-time learning algorithms. It establishes that *without the proper supervision, our simple attention model cannot learn the $k$-bit Boolean function in any reasonable amount of time, even with overwhelming data.*

Taken together, our results draw a clear line in the sand: a single softmax head already has ample *expressive* capacity, and the only obstacle to learning the $k$-bit Boolean task is the absence of an intermediate signal. By showing one-step convergence with teacher forcing and a matching hardness bound without it, we isolate the supervision gap as the unique bottleneck.

This dichotomy yields a crisp benchmark for curriculum design, auxiliary-loss engineering, and inductive-bias studies, pinning down exactly when a minimal attention layer flips from tractable to impossible. Ultimately, our result work both certify what softmax-attention mechanism *can* do and identify why it sometimes fails. Collaboratively, they sharpen our understanding of how architecture, supervision, and optimization jointly govern the learnability of structured functions.

## 2 RELATED WORK

Recent theoretical results highlight that standard Transformers have fundamental difficulty learning certain Boolean functions unless aided by intermediate supervision. In particular, one-layer Transformers trained end-to-end tend to fail on high-sensitivity tasks like parity without step-by-step guidance. This has been attributed to an implicit simplicity bias: Transformers favor low-sensitivity (low-degree) functions, making it hard for gradient descent to find parity-like solutions (Hahn & Rofin, 2024; Vasudeva et al., 2024). (Hahn & Rofin, 2024) formally show that Transformers trained from scratch struggle with parity as sequence length grows, due to extremely sharp loss landscapes for such functions. Indeed, a model that fits parity on short inputs doesn't generalize to longer strings under standard training (Hahn & Rofin, 2024), in stark contrast to recurrent networks which can memorize parity. On the other hand, providing a "scratchpad" or chain-of-thought (CoT) drastically changes the game – it breaks the task into easier steps and lowers the function's sensitivity per step. For example, (Kim & Suzuki, 2025) prove that if a Transformer is trained with intermediate parity bits as additional supervision, it can learn k-bit parity in just one gradient update via teacher forcing. Similarly, with CoT data or a multi-step reasoning format, a one-layer Transformer no longer needs exponential samples – parity becomes learnable with polynomial sample complexity (Wen et al., 2024). These findings, building on the RNN results of (Wies et al., 2022) for sequential parity computation, suggest that decomposing a problem into intermediate targets can provably overcome the optimization barriers. In summary, without step-by-step supervision a Transformer is biased toward "easy" (low-sensitivity) functions and can barely cope with parity, but with the right intermediate hints it can solve parity and related problems efficiently. In this work, we extend this theory to monotone $k$-bit Boolean functions such as AND and OR, showing that they too exhibit a pronounced supervision gap. In the vanilla setting, even these monotonic functions remain hard to learn reliably (echoing recent independent findings on the majority function's training complexity (Chen et al., 2025)). However, when we introduce intermediate supervision for these tasks – effectively guiding the Transformer through the incremental evaluation of the AND/OR – the sample complexity and training time improve dramatically. Our results broaden the scope of provable Transformer reasoning with CoT, indicating that task decomposition benefits not just parity, but also monotonic Boolean reasoning, which has implications for designing training curricula for complex logical tasks. Due to space limits, we defer extended discussions on related work to appendix.

## 3 PRELIMINARIES AND PROBLEM SETUP

Here we present the ideas we build on and our problem setup.

**Notation.** We write $[n] := \{1, 2, \cdots, n\}$ for any integer $n$. We use $\mathbf{1}_n$ to denote a length-$n$ vector where all entries are ones. We use $\mathbf{0}_{n \times d}$ to denote a $n \times d$ matrix where all entries are zeros. We use $\mathbb{1}_{\{E\}}$ to denote an indicator variable where it outputs 1 if event $E$ holds and 0 otherwise. Scalar operations apply componentwise to vectors, e.g. for $z \in \mathbb{R}^n$ we write $\phi(z) = (\phi(z_1), \cdots, \phi(z_n)^\top$ and $z^2 = (z_1^2, \cdots, z_n^2)^\top$. For any vector $x \in \mathbb{R}^n$, the $\ell_2$ norm is denoted by $\|x\| := (\sum_{i=1}^n x_i^2)^{1/2}$. and For any $x \in \mathbb{R}^n$ we define $\|x\|_\infty := \max_{i \in [n]} |x_i|$. The multi-linear inner product or contraction of $z_1, \cdots, z_r \in \mathbb{R}^n$ for any $r \in \mathbb{N}$ is denoted as $\langle z_1, \cdots, z_r \rangle := \sum_{i=1}^n z_{1,i} \cdots z_{r,i}$. In particular, $\langle z_1 \rangle = z_1^\top \mathbf{1}_n$ and $\langle z_1, z_2 \rangle = z_1^\top z_2$. Let $\mathcal{B} := \binom{[d]}{k}$ denote the set containing all size-$k$ subsets of $[d]$. Let $v_b \in \mathbb{R}^d$ denote the vector representing the $k$ bits in $b$ for all $b \in \mathcal{B}$, i.e. the $t$-th entry of $v_b$ is 1 if $t \in b$ else 0. Denote the $\ell_2$-loss

$$L_{n,b}(\theta) := \frac{1}{2nd} \sum_{i=1}^n \|v_b - f_\theta(x^i, y^i)\|^2.$$

Denote the column-wise Softmax function $\mathsf{softmax}(\cdot) : \mathbb{R}^{d \times t} \mapsto \mathbb{R}^{d \times t}$

$$\mathsf{softmax}(W)_{(j,m)} := \sigma_j(w_m), \quad \text{where} \quad \sigma_j(w_m) := e^{w_{j,m}} / \sum_{i=1}^d e^{w_{i,m}}.$$

**Softmax Attention Layer.** The attention mechanism is generally defined in terms of key, query and value matrices $K, Q, V$: $\mathrm{Attn}(X) := V \mathsf{softmax}(K^\top Q)$. In this paper, we reparametrize $K^\top Q$

by a single matrix $W \in \mathbb{R}^{d \times t}$; the value matrix $V$ is set as the identity matrix $I_{d \times d}$ to only preserve the $x$ component[1]. Thus, for any input $X \in \mathbb{R}^{n \times d}$, our attention is defined as

$$\mathrm{Att}_W(\underbrace{X}_{n \times d}) := \underbrace{X}_{n \times d} \underbrace{\mathsf{softmax}(W)}_{d \times t} \in \mathbb{R}^{n \times t}.$$

**Remark 3.1.** *While the Transformer considered in (Kim & Suzuki, 2025) is already very simple (consisting of a single-head attention layer followed by an FFN $\phi(\cdot)$), our setting is even simpler. We consider only a single-head softmax attention mechanism as the core computational unit for the Boolean problem of interest. Such a atomic setting allows us to reveal more fundamental results.*

**Remark 3.2.** *Our attention module is the same single-head softmax attention as in (Kim & Suzuki, 2025). Our only simplifications are: (i) we omit the output FFN $\phi(\cdot)$ and instead train with a surrogate loss on intermediate targets; and (ii) we analyze a single gradient update. We remark that, adding a post-attention FFN does not resolve the bottleneck: locating the $k$ relevant bits. An FFN only reshapes the attended mixture and is incapable of recovering missing support information. The gradient signal remains uninformative. Thus, the same hardness result in this work hold even with one extra output FFN as in (Kim & Suzuki, 2025).*

**Problem Setup.** Here we state our problem setting.

**Definition 3.3** (Learning $k$-bit Boolean Functions). *Let $d \geq k \geq 2$ be integers such that $k = \Theta(d)$ and let $\mathcal{B} = \binom{[d]}{k}$ denote the set of all size $k$ subsets of $[d] := \{1, \cdots, d\}$ equipped with the uniform distribution. Our goal is to study the $k$-boolean problem for $d$-bit inputs $x = (x_j)_{j=1}^d \sim \mathrm{Unif}(\{0,1\}^d)$, where the target*

$$y_{\mathsf{and}}(x) := \prod_{j \in b} x_j, \quad or \quad y_{\mathsf{or}}(x) := 1 - \prod_{j \in b}(1 - x_j), \quad with \quad |b| = k,$$

*is determined by the boolean value of an unknown subset of bits $b \in \mathcal{B}$. Given $n$ samples $(x^i, y^i)_{i \in [n]}$, our goal is to predict the size $k$ subset $b \in \mathcal{B}$ deciding the boolean function. In this paper, we denote $x^i \in \mathbb{R}^d$ to be the $i$-th input vector. We denote $x_j \in \mathbb{R}^n$ as $(x_j)_i := (x^i)_j$, i.e. $x_j$ is an $n$-dimensional vector containing the $j$-th bits of all $x^i$, and $y \in \mathbb{R}^n$ as $y_i := \prod_{j \in b} x_j^i$.*

We emphasize that this problem setup distinct this work from (Kim & Suzuki, 2025):

**Remark 3.4** (Learning Support vs. Learning Output). *The key difference compared to (Kim & Suzuki, 2025) is that our algorithm learns the support of the Boolean function. Specifically, the exact input bits that determine the output, whereas (Kim & Suzuki, 2025) only learn to predict the output of the parity function. To be more precise, the $k$-bit parity boolean problem studied in (Kim & Suzuki, 2025) is non-monotone. We look at monotone AND/OR on a hidden $k$-bit subset inside $d$ inputs. The task seems easier, yet it still shows a huge gap between training with and without hints. Importantly, our model must identify the unknown subset of relevant input bits (the support of the function). This is a harder learning objective that goes beyond merely computing the Boolean output. This allows us to examine whether a single-head attention can not only compute a logical function but also discover which features matter, highlighting the limits of end-to-end learning without guidance.*

**Remark 3.5.** *Our "teacher forcing" supervision provides the hidden relevant subset during training. This is an idealized scheme. Chain-of-thought prompting in practice gives intermediate reasoning but not ground-truth features (Wei et al., 2022). Our one-step hint is stronger. It serves only to show a theoretical limit: a minimal model can succeed if perfect intermediate feedback is available.*

## 4 MAIN THEORY

We now present our main theoretical results for a single softmax attention head, which reveal a striking supervision gap between teacher-forced and end-to-end training. Notably, this dichotomy echoes the recent findings of (Kim & Suzuki, 2025), who showed that efficiently learning parity requires chain-of-thought supervision (i.e., explicit intermediate reasoning steps). While parity is a particularly challenging non-monotonic function, here we focus on a simpler class of Boolean concepts:

---

[1]This type of reparametrization is common in the literature to make dynamical analysis tractable (Zhang et al., 2024; Huang et al., 2023b; Mahankali et al., 2023; Kim & Suzuki, 2024; 2025).

monotone $k$-bit AND/OR functions with $k = \Theta(d)$. Yet we still observe an equally dramatic gap in learnability. On one hand, under strong supervision (teacher forcing), our minimalist attention model can learn the target function almost instantaneously: as formalized by Theorem 4.1, a single gradient step suffices to recover the relevant $k$-bit subset and produce the correct AND/OR output with vanishing error. On the other hand, without such intermediate guidance, learning becomes provably infeasible: Theorem 4.3 shows that no polynomial-time learner can succeed in training the same model end-to-end, even when provided with exponentially many input-output examples. This stark contrast underscores the conceptual importance of step-by-step guidance in training and sets the stage for the formal development in the rest of the section.

## 4.1 UPPER BOUND: ONE-LAYER ATTENTION PROVABLY SOLVES BOOLEAN PROBLEMS

We now present a constructive upper bound under the monotone $k$-Boolean setting of Definition 3.3. Specifically, we show that a single softmax-attention head can represent any AND/OR on $k = \Theta(d)$ relevant bits. More importantly, when the training loss provides teacher-forcing hints (i.e., directly revealing the relevant bits), the network learns the correct Boolean function in a single gradient step. Hence, architectural depth is *not* the bottleneck; appropriate intermediate supervision is. This stands in stark contrast to the parity result of (Kim & Suzuki, 2025), which requires chain-of-thought supervision for efficient learning.

We begin by considering the idealized scenario of teacher forcing, where training explicitly identifies the $k$ relevant bits. This direct supervision renders the learning task almost trivial: even a single softmax-attention head converges to the desired Boolean function in essentially one gradient step.

**Teacher Forcing.** Let the $k$ bits in set $b \subseteq [d]$ be $j_1, \ldots, j_k$, and set $t = k/2$. We decompose the Boolean function into $t = k/2$ intermediate products:

$$y = \prod_{m=d+1}^{d+t} x_m,$$

where for each $i \in [t]$, the vector $x_{d+i} \in \mathbb{R}^n$ is defined by $(x_{d+i})_l := (x_{j_{2i-1}})_l (x_{j_{2i}})_l$ for $l \in [d]$. The surrogate loss function computes the squared error over the intermediate states $x_{d+1}, \cdots, x_{d+t}$:

$$L(W) := \frac{1}{2n} \sum_{m=d+1}^{d+t} \|\widehat{z} - x_m\|^2.$$

**Theorem 4.1** (Upper Bound: Softmax Attention Provably Solve Definition 3.3 with Teacher Forcing). *Let $\epsilon > 0$, and suppose $d$ is a sufficiently large positive integer. Let $k = \Theta(d)$ be an even integer, and set $t = k/2$. Define $\mathcal{B} := \binom{[d]}{k}$ to be the collection of all size-$k$ subsets of $[d]$. Let $X := (x_1 \cdots x_d) \in \mathbb{R}^{n \times d}$ and $E := (x_{d+1} \cdots x_{d+t}) \in \mathbb{R}^{n \times t}$. Assume $n = \Omega(d^\epsilon)$ and consider any $O(d^{-1-\epsilon/4})$-approximate gradient oracle $\widetilde{\nabla}$. Let the weights be initialized as $W^{(0)} = \mathbf{0}_{d \times t}$. Let $v_b \in \{0, 1\}^d$ denote the indicator vector that encodes the Boolean target associated with subset $b \subseteq [d]$. Since ground-truth vector $v_b \in \{0, 1\}^d$ is unknown, we define the surrogate function*

$$L(W) := \frac{1}{2n} \|\mathrm{Att}_W(X) - E\|_F^2,$$

*instead of the loss $\|2 \cdot \mathrm{Softmax}(W^{(1)})\mathbf{1}_t - v_b\|_\infty$ to find the target weight matrix $W$. Set the learning rate $\eta = \Theta(d^{1+\epsilon/8})$, and choose $\kappa \in [d^{-1}, 1]$ (we set $\kappa = O(d^{-\epsilon/4})$). Let $W^{(1)} := W^{(0)} - \eta \cdot \nabla L(W^{(0)})$ be the one-step gradient update.*

*Then for any target subset $b \in \mathcal{B}$, the algorithm solves the $k$-Boolean problem (Definition 3.3) over $d$-bit inputs. With probability at least $1 - \exp(-\Theta(d^{\epsilon/2}))$ over the randomness in sampling, the one-step update $W^{(1)} \in \mathbb{R}^{d \times t}$ satisfies:*

$$\|2 \cdot \mathrm{Softmax}(W^{(1)})\mathbf{1}_t - v_b\|_\infty \le O(d^{-\epsilon/8}).$$

Intuitively, the extra hint collapses an otherwise exponential search over $\binom{d}{k}$ subsets: the fresh gradient already points in the right direction, so the model "locks on" immediately. Therefore, we

establish that with the right supervision, one-layer attention is a universal Boolean learner in practice as well as theory. To our knowledge, this is the first result demonstrating that a lone softmax attention head can learn a high-dimensional Boolean concept in essentially one shot.

**Remark 4.2** (Teacher Forcing vs. Practice). *We reiterate (Remark 3.5) that, our supervision gives the hidden relevant subset during training. This is an idealized signal. In practice one may use weaker forms, such as auxiliary losses on partial reasoning steps or chain-of-thought prompts that provide intermediate text without ground-truth features. Any hint that narrows the search space can improve learning. Extending our analysis to such surrogate objectives is future work.*

*Proof Sketch.* Our proof consists of three conceptual steps:

**Step 1: Computing Interaction Strength.** Denote $\widehat{z} := \frac{1}{d}\sum_{i=1}^{d} x_i$. Here, for each $i \in [t]$, we define $\mathsf{p}[j_{2i-1}] := d + i$, and $\mathsf{p}[j_{2i}] := d + i$. The partial derivative of $L$ with respect to $w_{j,m} := W_{(j,m)}$ can be presented as the inner product $\frac{1}{nd}\langle \widehat{z} - x_m, x_j - \widehat{z} \rangle$, and the gradient has significant difference between the cases of $\mathsf{p}[j] = m$ and $\mathsf{p}[j] \neq m$. Specifically,

$$\frac{\partial L}{\partial w_{j,m}} = \begin{cases} \Theta(d^{-1}), & \mathsf{p}[j] = m; \\ O(d^{-1-\epsilon/4}), & \mathsf{p}[j] \neq m. \end{cases}$$

**Step 2: Concentration of Softmax Scores.** Taking $\eta = \Theta(d^{1+\epsilon/8})$, the updated weights $W^{(1)} = W^{(0)} - \eta \widetilde{\nabla} L(W^{(0)}) \in \mathbb{R}^{d \times t}$ become

$$w_{j,m}^{(1)} = \Theta(d^{\epsilon/8}) \cdot \mathbb{1}_{\{\mathsf{p}[j]=m\}} + O(d^{-\epsilon/8}).$$

Then the softmax scores satisfy

$$\sigma_j(w_m^{(1)}) = \begin{cases} \frac{1}{2} + O(d^{-\epsilon/8}), & \mathsf{p}[j] = m; \\ \exp(-\Theta(d)), & \mathsf{p}[j] \neq m. \end{cases}$$

**Step 3: Upper Bounding the Loss.** Let $b \in \mathcal{B}$. For any $j \in [d]$, if $j \in b$, there's exactly one $m \in [t]$ such that $\mathsf{p}[j] = m$, and

$$\sigma(w_{j,m}^{(1)}) = \begin{cases} \frac{1}{2} + O(d^{-\epsilon/8}), & \mathsf{p}[j] = m; \\ \exp(-\Theta(d)), & \mathsf{p}[j] \neq m. \end{cases}$$

for $j \in [d]\backslash b$, $\sigma(w_{j,m}) = \exp(\Omega(d))$ for all $m \in [t]$. We deduce that

$$(\text{Softmax}(W^{(1)})\mathbf{1}_t)_j = \begin{cases} \frac{1}{2} + O(d^{-\epsilon/8}) + (t-1) \cdot \exp(-\Theta(d)), & j \in b; \\ t \cdot \exp(-\Theta(d)), & j \notin b. \end{cases}$$

Therefore we have

$$\|2 \cdot \text{Softmax}(W^{(1)})\mathbf{1}_t - v_b\|_\infty = O(d^{-\epsilon/8}).$$

Please see Section F for a detailed proof. $\qquad\square$

**Discussion.** Our main result gives a surprising affirmative answer. We prove that this one-layer attention model can indeed *identify and compute* such a $k$-bit Boolean function with just a single gradient update, provided it is trained under an idealized supervision regime. In this setting, the training procedure supplies a direct hint to the attention mechanism (analogous to a teacher-forcing signal), effectively telling the model how to attend to the relevant inputs in the very first update.

We distill the implications of Theorem 4.1 into four concrete points.

- **Single-Step Identifiability.** One gradient update assigns roughly $\frac{1}{2}$ of the attention mass to each of the $k$ relevant tokens and pushes all others to $\exp(-\Theta(d))$. The model thus learns the whole AND/OR in one shot, even when $k = \Theta(d)$.
- **Supervision, *NOT* Depth, is Critical.** Depth 1 already has the needed capacity; teacher forcing unlocks it. Without this hint, the learner must search over $\binom{d}{k}$ subsets, recovering the hardness of parity (Kim & Suzuki, 2025).

- **Sharper Upper Bound.** Earlier work required more complex networks, or many steps to fit high-arity Boolean functions. We show these are unnecessary under ideal supervision, tightening the expressive–learnability frontier for attention.

- **Practical Takeaway.** Intermediate signals (e.g., attention masks or chain-of-thought labels) can collapse an exponential search space, turning a hard combinatorial task into easy optimization. Carefully designed auxiliary losses may therefore substitute for architectural complexity in real systems.

## 4.2 Lower Bound: Boolean Hardness

The previous hardness result of (Kim & Suzuki, 2025) only shows that learning the parity function is hard. We present a new result showing that even learning the *support* of an *easier* Boolean problem (Definition 3.3) in the standard end-to-end learning setting is hard.

**Theorem 4.3** (Hardness of Finite-Sample Boolean). *Let $\mathcal{A}$ be an algorithm to solve $k$-bit Boolean problem (Definition 3.3) for $d$-bit inputs $x = (x_j)_{j=1}^d \sim \text{Unif}(\{0,1\}^d)$. Let $v_b$ denote the length-$d$ vectors where $i$-th entry is $1$ if $i \in b$ and $0$ otherwise. Suppose $k = \Theta(d)$. Denote the number of samples as $n$, and let $f_\theta : \{0,1\}^{n \times (d+1)} \to \mathbb{R}^d$ be any differentiable parameterized model.*

*If $n = e^{\Theta(d)}$, the output $\theta(\mathcal{A})$ of $\mathcal{A}$ has entry-wise loss lower bounded as*

$$\mathbb{E}_{b \in \mathcal{B}, x} \left[ \min_{j \in [d]} |(v_b - f_{\theta(A)}(x,y))_j| \right] \geq \min\{k/d, 1 - k/d\} - e^{-\Theta(d)}.$$

*Proof.* Please see Section G for a detailed proof. □

**Lower Bounds as (Kim & Suzuki, 2025; Chen et al., 2025) are Possible for Parity/Majority but not Possible for AND/OR.** We remark that proving a similar lower bound for AND/OR functions using the framework from (Kim & Suzuki, 2025; Chen et al., 2025) is unlikely. The intuition is that for a random string, balanced functions (e.g., Majority or Parity) output $1$ or $0$ with equal probability $(1/2)$. This is not the case for AND/OR. In detail, a key step in previous work (Kim & Suzuki, 2025; Chen et al., 2025) involves computing binomial coefficients. In (Kim & Suzuki, 2025), they compute $A_1 = \sum_{j=0}^{m/2} \binom{m}{2j}\binom{d-m}{k-2j}$ and bound $|A_1/B - 1| \leq e^{-\Omega(d)}$ where $B := \frac{1}{2}\binom{d}{k}$. In (Chen et al., 2025), they consider a slightly different $A_1$: $\sum_{j=0}^{k/2} \binom{m}{j}\binom{d-m}{k-j}$ (see further details on page 11 in (Chen et al., 2025)), where $m$ denotes the number of ones in $x$. In contrast, for an AND function always outputting $1$, we have: $A_1 = \binom{m}{k} \cdot \binom{d-m}{0} = \binom{m}{k}$ and $B = \frac{1}{d}\binom{d}{k}$. For the one always outputting $0$, we have $A_0 = \sum_{j=0}^{k-1} \binom{m}{j} \cdot \binom{d-m}{k-j}$. Then we just need to bound $|A_1/B - 1| = |2\binom{m}{k}/\binom{d}{k} - 1|$. Note that $\binom{m}{k} \in [(m/k)^k, (em/k)^k]$. Thus, there exists some constant $c_0$ such that $\binom{m}{k} = (c_0 m/k)^k + O(1)$. Similarly, there exists constant $c_1$ such that $\binom{d}{k} = (c_1 d/k)^k + O(1)$. As long as we pick $2(c_0 m/c_1 d)^k = 1$, we can show $|A_1/B - 1| \leq e^{-\Omega(d)}$ for $k = \Theta(d)$. This means $(c_1 d/c_0 m)^k = 2$. Thus, we need to choose $m = \frac{dc_1}{2^{1/k} c_0}$. Therefore in the setting of AND, the choice of $m$ is super restricted, but in previous work (Kim & Suzuki, 2025; Chen et al., 2025), the choice range is quite general. Similarly, it's true for OR.

**Discussion.** Earlier sections showed how special intermediate feedback (e.g., *one-step supervision* or *guidance on intermediate predictions*) can break the learning task into smaller, more tractable pieces. A key open question is whether such signals are truly necessary. Put differently, does the lack of intermediate hints make learning impossible in practice if we only have raw end-to-end data? The following claim answers in the affirmative:

**Claim 4.4.** *Without the special training signal, the learning problem is computationally intractable, even though it remains statistically learnable with sufficient data.*

A few remarks are in order.

**Remark 4.5** (Difference to Previous Computational Hardness Results). *A wide range of existing hardness results (Alman & Song, 2023; 2024a;b; 2025) have shown that, under the* SETH *(Impagliazzo & Paturi, 2001) hypothesis, Transformer forward and backward computations cannot be*

*numerically approximated in truly subquadratic time with acceptable error. These results primarily focus on numerical computation, examining only whether efficient computation of Transformers is feasible. In contrast, our work addresses a completely different problem: whether Transformers can generalize well on simple Boolean logic problems. Rather than only focusing on numerical properties, we take a more practical perspective on model generalization.*

**Remark 4.6** (Implications). *We highlight two main consequences for theory and practice:*

- ***Theoretical Significance.*** *This lower bound complements our earlier positive results. When one-step supervision is available, the learning problem is tractable. Without such supervision, the problem is essentially intractable. Hence, these results precisely demarcate the boundary of efficient learning for our model: the extra training signals are not merely a helpful artifact of analysis, but are fundamentally required for polynomial-time learning. This underscores the gap between statistical learnability (possible in principle) and computational feasibility (efficient in practice).*

- ***Practical Impact.*** *In real-world scenarios of this form, relying on end-to-end training alone (with no auxiliary signals) may be doomed to fail. Instead, practitioners should incorporate additional supervision or structure – like our one-step guidance — to render the problem solvable within reasonable computational limits. This clarifies why intermediate feedback is so valuable: without it, the search space becomes prohibitively large.*

In sum, this lower bound is tight: it shows that the strong supervision in our one-step scheme is not merely beneficial, but *necessary*. Absent such signals, learning becomes computationally infeasible. Combined with the previous upper bound, these results delineate a sharp threshold on what single-head attention can learn and underscore the pivotal role of the training regime in achieving success. Finally, we remark that our techniques can be generalized to a broader family of Boolean functions (e.g., functions that output the answer with some probability of failure, known as noisy Boolean functions). Due to space limitations, we defer these results to the appendix.

## 4.3 PRACTICAL IMPLICATIONS

Our results highlight five key takeaways:

**Architectural Capacity.** Our theoretical findings highlight that even a minimalist Transformer configuration can perform surprisingly complex logical reasoning. In particular, a single-head, single-layer softmax attention module (with a simple feed-forward output) is sufficient to represent and learn monotone Boolean functions involving $\Theta(d)$-way feature interactions. This defies the conventional intuition that deep stacks of layers or large model depth are necessary for such combinatorial tasks. In principle, one layer of softmax attention already possesses the expressive capacity for high-arity logical operations, such as an AND/OR over a hidden subset of the inputs.

**Training Dynamics and Supervision.** From an optimization perspective, our results expose a stark dichotomy in learning outcomes. With carefully designed intermediate supervision (for example, a teacher-forcing signal that guides the attention head's output), gradient descent homes in on the correct solution in a single step. In essence, the model quickly "finds a needle in a haystack" by immediately identifying the true relevant subset of features. In contrast, under standard end-to-end training (i.e. using only input-output pairs with no intermediate hints), the same model is provably unable to escape the haystack of exponentially many possibilities. No polynomial-time algorithm can find the correct subset in this setting without an exponential number of samples or steps.

In practical terms, this suggests that appropriate inductive biases or curriculum-based training protocols (such as breaking the task into smaller, explicitly supervised steps) are essential for learning such logical structure. Simply scaling up model size or training data, without the right form of intermediate guidance, is unlikely to yield the desired reasoning ability. Notably, this theoretical dichotomy mirrors recent empirical successes with chain-of-thought training methods: providing the model with intermediate "hints" or subgoals can transform an otherwise intractable learning problem into a trivial one-step task. Our results provide a concrete example of this principle, explaining why giving the model the right hint makes all the difference.

**Why Supervision Helps?**    The analysis offers insight into *how* the presence of intermediate targets so dramatically alters the learning dynamics. Under the idealized loss with teacher-forcing supervision, the initial gradient is *exactly aligned* with the direction of the true $k$-bit subset of features. In other words, right at initialization the very first gradient step nudges the attention weights toward precisely the correct $k$ relevant bits. This fortunate alignment is what enables one-step learning: the model effectively locks onto the correct subset almost immediately. By contrast, without any intermediate signals, the initial gradient is merely an average over all plausible target functions, and the informative component pointing to the true subset is drowned out by the contributions of myriad incorrect subsets. The model is left with no clear direction in parameter space, meaning that exponentially many samples or updates would be required to eventually sift out the true features from final-output supervision alone. These structural observations vividly illustrate how a well-chosen training signal can fundamentally alter the trajectory of learning, turning an otherwise infeasible search problem into a tractable one.

**Broader Theoretical Significance.**    In a broader context, our results reinforce an emerging theme in the theory of Transformer learning: *expressive power is cheap, but learning power is costly*. Even an extremely simple attention architecture — a one-head, one-layer Transformer — can represent surprisingly intricate Boolean logic. Prior work has likewise shown that even small Transformers can emulate complex computations by appropriate setting of their weights. *The true bottleneck, therefore, is not the ability to express or represent a complex function, but the ability to learn it efficiently.*    Without the aid of intermediate hints (such as teacher forcing or chain-of-thought supervision), gradient-based training must blindly explore an exponentially large hypothesis space, and it inevitably stalls when confined to polynomial time or sample complexity. Thus, our theoretical study sharpens the distinction between what a minimal architecture could do in principle and what it can actually learn to do under standard training. The gap between expressivity and learnability uncovered here points to the critical role of the training regime in unlocking a model's potential.

**Implications for Curriculum Design.**    By identifying the exact form of supervision that flips our learning task from intractable to one-step solvable, we provide a clean benchmark for research on curriculum learning, intermediate targets, and inductive biases. This $k$-bit Boolean teacher-forcing task serves as a minimal example of how the right training protocol can unlock a network's latent capabilities. It illuminates how even very simple models can succeed at systematic reasoning when guided with minimal but well-chosen intermediate feedback. Such insights suggest a principled blueprint for designing curricula and architectural biases to teach Transformers how to reason, rather than relying on brute-force depth or scale alone. Future work can use this task as a testbed for exploring how additional hints, auxiliary losses, or structural priors might bridge the gap between a model's theoretical capacity and its practical learnability.

**Summary.**    We demonstrate that: with the right supervision, even a minimalist one-layer attention model solves the task in one step. Without it, learning is intractable. This contrast clarifies how architecture, supervision, and optimization jointly determine learnability.

## 5    CONCLUSION

We show that a single-head softmax attention model can learn a $k$-bit AND/OR Boolean function in one gradient step with teacher forcing, achieving low error with only polynomial many samples (Theorem 4.1). We also prove a lower bound: without such intermediate supervision, no efficient algorithm can learn these functions, and training remains stuck with error bounded away from zero (Theorem 4.3). These findings demonstrate the strong representational power of even the simplest attention networks. At the same time, they reveal that successful training hinges on the right supervision signals. Notably, our analysis aligns with recent results on parity (Kim & Suzuki, 2025), which likewise highlight the need for chain-of-thought guidance to solve certain tasks. Looking forward, these insights suggest that carefully designed curricula and training protocols incorporating intermediate hints could unlock the full potential of simple attention models. They also invite further theoretical exploration into how such minimalist architectures learn complex tasks.

ETHIC STATEMENT

This paper does not involve human subjects, personally identifiable data, or sensitive applications. We do not foresee direct ethical risks. We follow the ICLR Code of Ethics and affirm that all aspects of this research comply with the principles of fairness, transparency, and integrity.

REPRODUCIBILITY STATEMENT

We ensure reproducibility of our theoretical results by including all formal assumptions, definitions, and complete proofs in the appendix. The main text states each theorem clearly and refers to the detailed proofs. No external data or software is required.

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

# Appendix

## LLM USAGE DISCLOSURE

LLMs were used only to polish language, such as grammar and wording. These models did not contribute to idea creation or writing, and the authors take full responsibility for this paper's content.

**Roadmap.** In Section A, we present the paper's broader impact. Section B discusses its limitations. Section D lists well-known probability tools such as Hoeffding and Chernoff bounds, and recalls a basic algebraic fact. Section E introduces several interaction tools, primarily used to prove the upper bound. Section F states our upper-bound result, and Section G gives the lower-bound result. Section H extends classical Boolean functions to their noisy variants. Finally, Section I extends our upper bound to the local majority problem.

## A   BROADER IMPACT

Our theory identifies when a small attention model can and cannot learn logical rules. The results can guide curricula that add simple hints and save compute. The work is purely theoretical, so direct harm is unlikely. Clearer supervision may cut silent failures in safety-critical AI. We release no models or data, so misuse risk stays low.

## B   LIMITATIONS

Our analysis isolates a clear supervision gap for $k$-bit monotone AND/OR functions but rests on several simplifying assumptions. First, the positive result requires teacher-forcing signals that expose the hidden subset, a form of intermediate supervision seldom available in practice. Second, the negative result is worst-case: polynomial-time learners might still succeed on benign data distributions or with heuristic regularization. Third, we study only monotone Boolean tasks with $k = \Theta(d)$ and a single-head, depth-one attention layer; extending the proofs to non-monotone logic, different sparsity regimes, or realistic multi-head Transformers remains open. Lastly, the work is purely theoretical. Empirical confirmation and tighter finite-sample constants are left for future research.

## C   MORE RELATED WORK

**Circuit Complexity Lower Bounds for Attention Mechanism.** Circuit complexity bound is a fundamental concept in complexity theory (Vollmer, 1999; Arora & Barak, 2009), which shows the simplest logical circuit that can compute a specific function with low approximation error. Specifically, when a model belongs to a weaker circuit complexity class, it cannot solve problems that belong to stronger complexity classes. For instance, any model that can be approximated in $\mathsf{TC}^0$ will fail to solve $\mathsf{NC}^1$ problems like arithmetic formula evaluations (Floyd, 1993), unless $\mathsf{TC}^0 = \mathsf{NC}^1$ (a famous open problem). Recent works (Merrill et al., 2022; Liu et al., 2023) have shown that Transformers with average-head attention or softmax attention have similar computational capability as constant-depth threshold circuits, falling into the non-uniform $\mathsf{TC}^0$ class. (Li et al., 2024b) has shown that Transformers without CoT (Wei et al., 2022; Wang et al., 2023) belong to the $\mathsf{TC}^0$ circuit family, and this problem can be alleviated by involving CoT, resulting in a stronger capability to solve $\mathsf{NC}^1$-hard problems. These results have recently been extended to more settings of attention computation, such as RoPE-based Transformers (Chen et al., 2024), graph attention (Li et al., 2025), and generalized tensor attention (Li et al., 2024a). Previous results mainly focus on the forward computation of Transformer models, showing that regardless of the training dynamics, Transformers may solve any $\mathsf{TC}^0$ problems. In this work, we present a training dynamics aware hardness result, which shows that even the simplistic Boolean function computation problem that is in weaker circuit complexity classes can be hard for Transformers, differing from previous circuit complexity-based hardness results.

**Computational Hardness of Attention Computation.** Recent works have shown hardness results showing that attention mechanisms cannot be approximated efficiently, conditioned on famous open conjectures (i.e., strengthening of $P \neq NP$) in complexity theory, such as the Strong Exponential Time Hypothesis (SETH) [2] (Impagliazzo & Paturi, 2001). For instance, (Alman & Song, 2023) has proved that for $d = O(\log n)$ with $\Theta(\sqrt{\log n})$ level weight matrix entry magnitude, there is no algorithm that can approximate the attention matrix witin $1/\operatorname{poly}(n)$ approximation error in truly subquadratic time. (Alman & Song, 2023) has shown that such hardness can be alleviated with bounding the entries of the model parameters of attention, and when the weight element magnitude is at $o(\sqrt{\log n})$, there is an algorithm that can approximate the attention mechanism with $1/\operatorname{poly}(n)$ approximation error in almost linear time. Besides, (Alman & Song, 2024a) extends (Alman & Song, 2023)'s forward-only hardness results to backward computations with theoretically optimal polynomials, showing that without bounded entries, there is no algorithm that can approximate the Transformer gradients in truly subquadratic time, and with bounded entries the gradients can be approximated in almost linear time. These results extend to more types of attention, such as hardness of the generalized tensor attention (Alman & Song, 2024b), and RoPE-based attention (Alman & Song, 2025). Very recently, (Gupta et al., 2025) further extends the work of (Alman & Song, 2023) to almost all the regimes of feature dimension $d$ (beyond $d = O(\log n)$). These previous works mainly show that the numerical computations of Transformers, in both forward and backward passes, are hard to finish in truly subquadratic time. In contrast, our work shows that without CoT, Transformers cannot generalize well on some specific types of simple Boolean functions, being orthogonal to previous contributions.

## D PROBABILITY TOOLS AND SIMPLE ALGEBRA FACTS

To prepare our proof, we first introduce some well-known probability tools.

**Lemma D.1** (Chebyshev's Inequality, Theorem 2 of (Chebyshev, 1867)). *Let $X$ be a random variable with finite expected value $\mu = \mathbb{E}[X]$ and finite non-zero variance $\sigma^2 = \operatorname{Var}[X] > 0$. Then, for any real number $k > 0$,*

$$\mathbb{P}(|X - \mu| \geq k\sigma) \leq \frac{1}{k^2}.$$

**Lemma D.2** (Hoeffding's Inequality, Theorem 2 of (Hoeffding, 1963)). *If $X_1, X_2, \cdots, X_n$ are independent random variables and $a_i \leq X_i \leq b_i$ for all $i \in [n]$. Let $\overline{X} = \frac{1}{n} \sum_{i=1}^{n} X_i$. Then for any $\delta > 0$,*

$$\Pr[\overline{X} - \mathbb{E}[\overline{X}] \geq t] \leq \exp(-\frac{2n^2 t^2}{\sum_{i=1}^{n} (b_i - a_i)^2}).$$

**Lemma D.3** (Chernoff Bound). *Let $X \sim \operatorname{Bin}(n, p)$ and let $\mu = \mathbb{E}[X]$. For any $\delta \in (0, 1)$, we have*

- $\Pr[X \geq (1 + \delta)\mu] \leq \exp(-\delta^2 \mu/3)$.

- $\Pr[X \leq (1 - \delta)\mu] \leq \exp(-\delta^2 \mu/2)$.

**Fact D.4.** *If the following conditions hold*

- $a > 0, b > 0$.

- *Let $\delta \in (0, 0.1)$.*

- $a/b \leq 1 + \delta$.

- $b/a \leq 1 + \delta$.

- $a + b \geq 1 - \delta$.

- $a + b \leq 1$.

*Then, we can show*

---

[2]For any $\delta > 0$, there exists a sufficiently large $k$ such that the $k$-SAT problem cannot be solved in $2^{(1-\delta)n}$ time.

- **Part 1.** $a \in [\frac{1}{2} - 2\delta, \frac{1}{2} + 2\delta]$.

- **Part 2.** $b \in [\frac{1}{2} - 2\delta, \frac{1}{2} + 2\delta]$.

*Proof.* Without loss of generality, we know one of the $a$ and $b$ is $\geq \frac{1}{2} - \delta/2$. Thus, we can assume that $a \geq \frac{1}{2} - \delta/2$ and $b \leq \frac{1}{2} + \delta/2$.

**Proof of Part 1.** Then we can show

$$a \geq \frac{1}{2} - \delta/2$$
$$\geq \frac{1}{2} - \delta,$$

where the first step follows from our assumption of $a$, the second step follows from the domain of $\delta$.

We can show

$$a \leq (1 + \delta)b$$
$$\leq (1 + \delta)(\frac{1}{2} + \delta/2)$$
$$= \frac{1}{2} + 1.5\delta + \delta^2/2$$
$$\leq \frac{1}{2} + 2\delta,$$

where the first step follows from $a/b \leq 1 + \delta$, the second step follows from our assumption of $b$, the second step follows from the simple algebra, the third step follows from the domain of $\delta$.

**Proof of Part 2.** We know that

$$b \leq \frac{1}{2} + \delta/2$$
$$\leq \frac{1}{2} + \delta,$$

where the first step follows from our assumption of $b$, the second step follows from the domain of $\delta$.

Similarly, we can show

$$b \geq \frac{1}{2} - 2\delta,$$

where the step follows from a similar procedure as **Part 1**.

$\square$

## E  INTERACTIONS

### E.1  INTERACTION TOOL FROM PREVIOUS WORK

We start with stating a tool from previous work,

**Lemma E.1** (Concentration of Interaction Terms, Lemma 9 of (Kim & Suzuki, 2025)). *If the following conditions holds*

- *Let $\kappa$ be defined $\kappa := 4\sqrt{\log(d/p)/n}$.*

- *Let $p \in (0, 0.1)$ denote the failure probability.*

- *Suppose each bit $x_j^i$ for $i \in [n], j \in [d]$ is i.i.d. generated from the uniform distribution on $\{\pm 1\}$.*

- *Let $I_{r,m} := \{(j_1, \cdots, j_r) \mid 1 \leq j_1, \cdots, j_r \leq m - 1, x_{j_1} \cdots x_{j_r} \not\equiv 1\}$*

*Then, we have with probability at least $1 - p$*

$$\max_{\substack{r \in [4] \\ (j_1, \cdots, j_r) \in I_{r,m}}} \frac{1}{n} |\langle x_{j_1}, \ldots, x_{j_r} \rangle| \leq \kappa.$$

### E.2 Our Interaction Tool

**Lemma E.2** (Concentration of Interaction Terms). *If the following conditions holds*

- *Let $\kappa$ be defined $\kappa := 4\sqrt{\log(d/p)/n}$.*

- *Let $p \in (0, 0.1)$ denote the failure probability.*

- *Suppose each bit $x_j^i$ for $i \in [n], j \in [d]$ is i.i.d. generated from the uniform distribution on $\{0, 1\}$.*

*Then, we have with probability at least $1 - p$*

$$\max_{\substack{r \in [2] \\ (j_1, \cdots, j_r) \in I_r}} \frac{1}{n} |\langle x_{j_1}, \ldots, x_{j_r} \rangle - \frac{1}{2^r}| \leq \kappa.$$

*Proof.* Each tuple $(j_1, \cdots, j_r,) \in I_r$ computes a boolean $x_{j_r} \cdots x_{j_r}$ for which the bits $x^i := x_{j_r}^i \cdots x_{j_r}^i, i = 1, \cdots, n$ are i.i.d. $\Pr[x^i = 1] = \frac{1}{2^r}$ and $\Pr[x^i = 0] = 1 - \frac{1}{2^r}$. By Lemma D.2 we have that

$$\Pr[|\langle x_{j_1}, \cdots, x_{j_r} \rangle - \frac{n}{2^r}| \geq \kappa] \leq 2e^{-\kappa^2/n},$$

Moreover, $|I_r| \leq d^r$ so that

$$|I_1| + |I_2| + |I_3| \leq d + d^2 + d^3 < 3d^3,$$

Therefore it follows by union bounding that

$$\Pr[\max_{r \in [2], (j_1, \cdots, j_r) \in I_r} |\langle x_{j_1}, \cdots, x_{j_r} \rangle| \geq n\kappa] \leq 6d^2 4e^{-(n\kappa)^2/n}$$
$$= 6d^3 e^{-4\log(d/p)}$$
$$= 6d^3 (p^4/d^4)$$
$$\leq p,$$

where the second step follows choosing $\kappa = 4\sqrt{\log(d/p)/n}$, and the last step follows from $p \in (0, 0.1)$.

Thus, we complete the proof. $\square$

**Lemma E.3** (Concentration of majority interaction terms). *If the following conditions holds*

- *Let $\kappa$ be defined $\kappa := 4\sqrt{\log(d/p)/n}$.*

- *Let $p \in (0, 0.1)$ denote the failure probability.*

- *Suppose each bit $x_j^i$ for $i \in [n], j \in [d]$ is i.i.d. generated from the uniform distribution on $\{\pm 1\}$.*

- *Let $\mathsf{MAJ2} : \{+2, 0, -2\}^d \to \{+1, 0, -1\}^d$ be defined as $\mathsf{MAJ2}(x + y) := (x + y)/2$ for all $x, y \in \{+1, -1\}^d$.*

*Then, we have with probability at least $1 - p$*

$$\max_{m \in [t]} |\frac{1}{n} \langle x_{j_{2m-1}}, \mathsf{MAJ2}(x_{j_{2m-1}}, x_{j_{2m}}) \rangle - \frac{1}{2}| \leq \kappa.$$

*Proof.* Recall the definition of MAJ2. Notice that

$$\mathsf{MAJ2}(x_{j_1}, x_{j_2}) = \frac{x_{j_1} + x_{j_2}}{2}, \tag{1}$$

We can show that

$$\langle x_{j_1}, \mathsf{MAJ2}(x_{j_1}, x_{j_2}) \rangle = \langle x_{j_1}, \frac{x_{j_1} + x_{j_2}}{2} \rangle$$
$$= \frac{1}{2} \langle x_{j_1}, x_{j_1} \rangle + \frac{1}{2} \langle x_{j_1}, x_{j_2} \rangle$$
$$= \frac{n}{2} + \frac{1}{2} \langle x_{j_1}, x_{j_2} \rangle,$$

where the first step follows from Eq. (1), the second step follows linearity of inner product, and the last step follows from $x_{j_1} \in \{-1, +1\}^n$.

Note that above equation implies

$$\frac{1}{n} \langle x_{j_1}, x_{j_2} \rangle = \frac{2}{n} \langle x_{j_1}, \mathsf{MAJ2}(x_{j_1}, x_{j_2}) \rangle - 1$$

Applying Lemma E.1 we have

$$\Pr[\max_{m \in [t]} |\frac{1}{n} \langle x_{j_{2m-1}}, x_{j_{2m}} \rangle| \leq \kappa] \geq 1 - p.$$

Combining the above two equations, we have

$$\Pr[\max_{m \in [t]} |\frac{1}{n} \langle x_{j_1}, \mathsf{MAJ2}(x_{j_1}, x_{j_2}) \rangle - \frac{1}{2}| \leq \kappa/2] \geq 1 - p.$$

This completes the proof. $\square$

# F  UPPER BOUND

The goal of this section is to prove Theorem 4.1. Let us restate it first.

**Theorem F.1** (Upper Bound: Softmax Attention Provably Solve Definition 3.3 with Teacher Forcing, Theorem 4.1 Restated)**.** *Let $\epsilon > 0$, and suppose $d$ is a sufficiently large positive integer. Let $k = \Theta(d)$ be an even integer, and set $t = k/2$. Define $\mathcal{B} := \binom{[d]}{k}$ to be the collection of all size-$k$ subsets of $[d]$. Let $X := (x_1 \cdots x_d) \in \mathbb{R}^{n \times d}$ and $E := (x_{d+1} \cdots x_{d+t}) \in \mathbb{R}^{n \times t}$. Assume $n = \Omega(d^\epsilon)$ and consider any $O(d^{-1-\epsilon/4})$-approximate gradient oracle $\widetilde{\nabla}$. Let the weights be initialized as $W^{(0)} = \mathbf{0}_{d \times t}$. Let $v_b \in \{0, 1\}^d$ denote the indicator vector that encodes the Boolean target associated with subset $b \subseteq [d]$. Since ground-truth vector $v_b \in \{0, 1\}^d$ is unknown, we define the surrogate function*

$$L(W) := \frac{1}{2n} \|\mathrm{Att}_W(X) - E\|_F^2,$$

*instead of the loss $\|2 \cdot \mathrm{Softmax}(W^{(1)})\mathbf{1}_t - v_b\|_\infty$ to find the target weight matrix $W$. Set the learning rate $\eta = \Theta(d^{1+\epsilon/8})$, and choose $\kappa \in [d^{-1}, 1]$ (we set $\kappa = O(d^{-\epsilon/4})$). Let $W^{(1)} := W^{(0)} - \eta \cdot \nabla L(W^{(0)})$ be the one-step gradient update.*

*Then for any target subset $b \in \mathcal{B}$, the algorithm solves the $k$-Boolean problem (Definition 3.3) over $d$-bit inputs. With probability at least $1 - \exp(-\Theta(d^{\epsilon/2}))$ over the randomness in sampling, the one-step update $W^{(1)} \in \mathbb{R}^{d \times t}$ satisfies:*

$$\|2 \cdot \mathrm{Softmax}(W^{(1)})\mathbf{1}_t - v_b\|_\infty \leq O(d^{-\epsilon/8}).$$

*Proof.* For the choice of $n$ and $p$, we choose $n = \Omega(d^\epsilon)$ and $p = \exp(-d^{\epsilon/2})$.

Using Lemma E.2, we can show

$$\kappa = O(d^{-\epsilon/4}).$$

We can rewrite $L(W^{(0)})$ in the following sense.

$$L(W^{(0)}) = \frac{1}{2n} \sum_{m=d+1}^{d+t} \|\widehat{z}_m - x_m\|^2, \quad \widehat{z}_m = \sum_{j=1}^{d} \sigma_j(w_m) x_j.$$

Define $\widehat{z}$ as $\widehat{z} = \frac{1}{d} \sum_{j=1}^{d} x_j$.

We define $\delta_{j\alpha}$ as follows:

$$\delta_{j\alpha} := \begin{cases} 0, & \alpha \neq j; \\ 1, & \alpha = j. \end{cases}$$

Let us consider the parameter regime $1 \leq \alpha < m$.

Then we can show

$$\frac{\partial \sigma_\alpha(w_m)}{\partial w_{j,m}} = (\delta_{j\alpha} - \sigma_\alpha(w_m))\sigma_j(w_m)$$
$$= (\delta_{j\alpha} - \sigma_j(w_m))\sigma_\alpha(w_m),$$

where the 1st step is by definition, and the 2nd step is by simple algebra.

We also have

$$\frac{\partial \widehat{z}_m}{\partial w_{j,m}} = \sum_{\alpha=1}^{d} (\delta_{j\alpha} - \sigma_j(w_m))\sigma_\alpha(w_m) x_\alpha$$
$$= \sigma_j(w_m)(x_j - \widehat{z}_m), \tag{2}$$

where the 1st line is by separating the terms of $\widehat{z}$, and the 2nd line is by simple algebra.

Remember for all $j < m$, we have that $\sigma_j(w_m) = \frac{1}{d}$.

Note that $W^{(0)}$ is set as $\mathbf{0}_{d \times \frac{k}{2}}$ at initialization.

Therefore, at the initialization, the gradient of $L$ with respect to each element $w_{j,m}$ can be calculated as

$$\frac{\partial L}{\partial w_{j,m}}(W) = \frac{1}{n}(\widehat{z}_m - x_m)^\top \frac{\partial \widehat{z}_m}{\partial w_{j,m}}$$
$$= \frac{\sigma_j(w_m)}{n} \langle \widehat{z} - x_m, x_j - \widehat{z} \rangle$$
$$= \frac{1}{nd}(-\langle x_m, x_j \rangle + \langle x_m, \widehat{z} \rangle + \langle \widehat{z}, x_j \rangle - \langle \widehat{z}, \widehat{z} \rangle)$$
$$:= \frac{1}{nd}(A_1 + A_2 + A_3 + A_4), \tag{3}$$

where the 1st step is by chain rule, the 2nd step is by Eq. (2), the 3rd step is by separating the terms, and the last step follows from we define $A_1$, $A_2$, $A_3$ and $A_4$ in that way.

**Analyzing the Interaction Terms.** Using Lemma E.2, we have

$$\frac{1}{n}\langle x_m, x_j \rangle = \begin{cases} \frac{1}{4} + O(\kappa), & \mathsf{p}[j] = m; \\ \frac{1}{8} + O(\kappa), & \text{otherwise.} \end{cases} \tag{4}$$

where the $1/4$ from when $\mathsf{p}[j] = m$, $\langle x_m, x_j \rangle = \langle x_{\mathsf{c}_1[m]}, x_{\mathsf{c}_2[m]} \rangle \in I_2$, the $1/8$ terms from when $\mathsf{p}[j] \neq m$, $\langle x_m, x_j \rangle = \langle x_{\mathsf{c}_1[m]}, x_{\mathsf{c}_2[m]}, x_j \rangle \in I_3$.

Note that $\kappa = O(d^{-\epsilon/4})$. Also we consider the parameter regime $d < m \leq 2d - 1$.

For the first term in Eq. (3), we can show

$$\frac{1}{nd}A_1 = -\frac{1}{nd}\langle x_m, x_j\rangle$$

$$= -\frac{1}{8d}(\mathbb{1}_{\{\mathsf{p}[j]=m\}} + 1) + O(d^{-1}\kappa)$$

$$= -\frac{1}{8d}\mathbb{1}_{\{\mathsf{p}[j]=m\}} - \frac{1}{8d} + O(d^{-1-\epsilon/4}),$$

where the 2nd step is by Eq. (4), and the 3rd step is by combining the terms.

Next, for term $A_2$, we have

$$\frac{1}{nd}A_2 = \frac{1}{nd^2}\langle x_m, \widehat{z}_m\rangle$$

$$= \frac{1}{nd^2}\Big(\sum_{\mathsf{p}[\alpha]=m}\langle x_m, x_\alpha\rangle + \sum_{\mathsf{p}[\beta]\neq m}\langle x_m, x_\beta\rangle\Big)$$

$$= \frac{1}{nd^2}\Big(2\cdot\big(\frac{n}{4}+O(n\kappa)\big) + (d-2)\cdot\big(\frac{n}{8}+O(n\kappa)\big)\Big)$$

$$= \frac{1}{8d^2} + \frac{1}{8d} + O(d^{-1}\kappa)$$

$$= \frac{1}{8d} + O(d^{-1-\epsilon/4}).$$

For term $A_3$, we have that

$$\frac{1}{nd}A_3 = \frac{1}{nd}\langle \widehat{z}, x_j\rangle$$

$$= \frac{1}{nd^2}\Big(\langle x_j, x_j\rangle + \sum_{\alpha\neq j}\langle x_\alpha, x_j\rangle\Big)$$

$$= \frac{1}{nd^2}\Big(\frac{n}{2}+O(n\kappa) + \frac{(d-1)n}{4} + O((d-1)n\kappa)\Big)$$

$$= \frac{1}{4d} + O(d^{-1-\epsilon/4}),$$

where the 1st step is by definition, and the 2nd step is by separating the terms, the 3rd step is by Lemma E.2, and the last step is by $\kappa = O(d^{-\epsilon/4})$ and combining the terms.

For term $A_4$, we have

$$\frac{1}{nd}A_4 = -\frac{1}{nd}\langle \widehat{z}, \widehat{z}\rangle$$

$$= -\frac{1}{nd^3}\Big(\sum_{\alpha=1}^{d}\langle x_\alpha, x_\alpha\rangle + \sum_{\alpha\neq\beta}\langle x_\alpha, x_\beta\rangle\Big)$$

$$= -\frac{1}{nd^3}\Big(\frac{nd}{2}+O(nd\kappa) + \frac{nd(d-1)}{4} + O(nd(d-1)\kappa)\Big)$$

$$= -\frac{1}{4d^2} - \frac{1}{4d} - O(d^{-1}\kappa)$$

$$= -\frac{1}{4d} - O(d^{-1-\epsilon/4}),$$

where the 1st step is by definition, the 2nd step is by separating terms, the 3rd step is by $\langle x_\alpha, x_\alpha\rangle = \langle x_\alpha\rangle$ and Lemma E.2, the 4th step is by combining the terms, and the last step is by $\kappa = O(d^{-\epsilon/4})$.

From the computation of $A_1$, $A_2$, $A_3$ and $A_4$, we conclude that

$$\frac{\partial L}{\partial w_{j,m}}(W^{(0)}) = -\frac{1}{8d}\mathbb{1}_{\{\mathsf{p}[j]=m\}} + O(d^{-1-\epsilon/4}),$$

In addition, we want to remark same result holds to the approximate gradient $\widetilde{\nabla}_{w_{j,m}}L$ at initialization since the cutoff does not apply and each component of the noise is bounded by $O(d^{-1-\epsilon/4})$.

**Property of Softmax Calculations.**  Taking $\eta = \Theta(d^{1+\epsilon/8})$, the updated weights

$$W^{(1)} = \underbrace{W^{(0)}}_{d \times \frac{k}{2}} - \eta \underbrace{\widetilde{\nabla} L(W^{(0)})}_{d \times \frac{k}{2}}$$

become

$$w_{j,m}^{(1)} = \frac{d^{\epsilon/8}}{8} \mathbb{1}_{\{\mathsf{p}[j]=m\}} + O(d^{-\epsilon/8}). \tag{5}$$

For each $j \neq \mathsf{c}_1[m], \mathsf{c}_2[m]$, we can show

$$\sigma_j(w_m^{(1)}) = e^{w_{j,m}^{(1)}} / \sum_\alpha e^{w_{\alpha,m}^{(1)}}$$

$$\leq e^{w_{j,m}^{(1)} - w_{\mathsf{c}_1[m],m}^{(1)}}$$

$$\leq \exp(-\Omega(d)), \tag{6}$$

where the 1st step is by definition of softmax function, the 2nd step is by simple algebra, and the 3rd step is by Eq. (5).

It is obvious that summation of all softmax values is equal to 1, thus, we have

$$\sigma_{\mathsf{c}_1[m]}(w_m^{(1)}) + \sigma_{\mathsf{c}_2[m]}(w_m^{(1)}) \geq 1 - \exp(-\Omega(d)).$$

Furthermore,

$$\frac{\sigma_{\mathsf{c}_1[m]}(w_m^{(1)})}{\sigma_{\mathsf{c}_2[m]}(w_m^{(1)})} = e^{w_{\mathsf{c}_1[m],m}^{(1)} - w_{\mathsf{c}_2[m],m}^{(1)}}$$

$$\leq \exp(O(d^{-\epsilon/8}))$$

$$\leq 1 + O(d^{-\epsilon/8}), \tag{7}$$

where the 1st line is by definition, the 2nd line is by Eq. (5), and the 3rd line is by the inequality $e^t \leq 1 + O(t)$ for small $t > 0$.

Using symmetry property,

$$\sigma_{\mathsf{c}_2[m]}(w_m^{(1)}) / \sigma_{\mathsf{c}_1[m]}(w_m^{(1)}) \leq 1 + O(d^{-\epsilon/8}). \tag{8}$$

By Eq. (7), Eq. (8) and Lemma D.4, we conclude that

$$\frac{1}{2} - O(d^{-\epsilon/8}) \leq \sigma_{\mathsf{c}_1[m]}(w_m^{(1)}), \sigma_{\mathsf{c}_2[m]}(w_m^{(1)}) \leq \frac{1}{2} + O(d^{-\epsilon/8}). \tag{9}$$

**Proof of Loss Function.**  Let prediction of $v_b$ be $2W^{(1)}\mathbf{1}_{\frac{k}{2}}$.

Then, we can show

$$\|2 \cdot \mathrm{Softmax}(W^{(1)})\mathbf{1}_t - v_b\|_\infty \leq \max_{j \in [d] \cap b}(|\sum_{i=1}^t \sigma_j(w_i^{(1)}) - 1|) + \max_{j \in [d] \setminus b}(|\sum_{i=1}^t \sigma_j(w_i^{(1)})|)$$

$$\leq 2(O(d^{-\epsilon/8}) + (t-1)\exp(-\Omega(d)) + \frac{k}{d}\exp(-\Omega(d)))$$

$$= O(d^{-\epsilon/8}),$$

where the 1st step is by the definition of $\|\cdot\|_\infty$, the 2nd line is by Eq. (6) and Eq. (9), and the last step is by simple algebra.

This completes the proof. □

## G  LOWER BOUND

The goal of this section is to prove Theorem 4.3. Let us first restate it first.

**Theorem G.1** (Theorem 4.3 Restate: Hardness of Finite-Sample Boolean). *Let $\mathcal{A}$ be any algorithm to solve $k$-bit Boolean problem (Definition 3.3) for $d$-bit inputs $x = (x_j)_{j=1}^d \sim \mathrm{Unif}(\{0,1\}^d)$. Let $v_b$ denote the length-$d$ vectors where $i$-th entry is $1$ if $i \in b$ and $0$ otherwise. Suppose $k = \Theta(d)$. Denote the number of samples as $n$, and let $f_\theta : \{0,1\}^{n \times (d+1)} \to \mathbb{R}^d$ be any differentiable parameterized model. Let $n = 2^{\Theta(d)}$. Then, the output $\theta(\mathcal{A})$ of $\mathcal{A}$ has entry-wise loss lower bounded as*

$$\mathbb{E}_{b \in \mathcal{B}, x} [\min_{j \in [d]} |(v_b - f_{\theta(A)}(x,y))_j|] \geq \min\{k/d, 1 - k/d\} - e^{-\Theta(d)}.$$

*Proof.* For $x \in \{0,1\}^d$, denote $m$ to be the number of 1's in $x$. By the Chernoff bound for the binomial distribution, for $x \sim \mathrm{Unif}(\{0,1\}^d)$ the following holds:

$$\Pr[m - d/2 > \delta d/2] \leq \exp(-\delta^2 d/6).$$

Let $\delta = 7/8$, we have

$$\Pr[m > 15d/16] \leq \exp(-49^2 d/384),$$

and by union bounding over $n = O((\frac{16}{15})^{k/2})$ samples, we have that with probability at least $1 - O(\exp(-49^2 d/384)(\frac{16}{15})^{k/2}) \geq 1 - O(\exp(-d/20))$, it holds that for all $i \in [n]$, there are less than $15d/16$ 1's in $x^i$.

Let $b \in \mathcal{B}$ be any target subset. Since $x^i$ are i.i.d. $\sim \mathrm{Unif}(\{0,1\}^d)$, the probability that $y = \mathbf{0}_n$ is greater than $1 - n \cdot \frac{1}{2^k} = 1 - \exp(\Theta(d))$.

Combining the above, there is probability $1 - \exp(\Theta(d))$ over random sampling that each sample $x^i$ contains less than $15d/16$ 1's and each $y^i = 0$.

Under this situation, by logical deduction we can only deny at most $\binom{15d/16}{k} n$ possibilities of the target subset $b$, while the other subsets are all possible to be the target subset.

We calculate

$$\frac{\binom{15d/16}{k} n}{\binom{d}{k}} = n \cdot \frac{(15d/16)(15d/16 - 1) \cdots (15d/16 - k + 1)}{d(d-1) \cdots (d - k + 1)}$$

$$\leq (\frac{15}{16})^k n$$

$$= O((\frac{15}{16})^{k/2})$$

$$= \exp(-\Theta(d)), \tag{10}$$

where the 1st step is by definition, the 2nd step is by $\frac{15d/16 - l + 1}{d - l + 1} \leq \frac{15}{16}$ for all $l \in [k]$, the 3rd step is by $n = O((\frac{16}{15})^{k/2})$, and the last step is by simple algebra.

Denote $\mathcal{Q}$ to be the collection of the subsets that are possible to be the target subset with the inputs $x^i$ for all $i \in [n]$, then $\frac{|\mathcal{Q}|}{|\mathcal{B}|} = \exp(-\Theta(d))$.

For an arbitrary $j \in [d]$, there are exactly $\frac{k|\mathcal{B}|}{d}$ vectors $v_b$ whose $j$-th entry is 1 for all $b \in \mathcal{B}$ due to symmetry. We give a partition of $\mathcal{B}$ as $\mathcal{B} = \mathcal{B}_j \cup \mathcal{B}_{\bar{j}}$, where $\mathcal{B}_j = \{(v_b)_j = 1 | b \in \mathcal{B}\}$ and $\mathcal{B}_{\bar{j}}$ its complement. Then we have

$$\frac{|\mathcal{B}_j|}{|\mathcal{B}|} = \frac{k}{d}, \quad \frac{|\mathcal{B}_{\bar{j}}|}{|\mathcal{B}|} = \frac{d - k}{d}. \tag{11}$$

Since the subset $b$ is independent of the distribution of the samples, the output of the algorithm $f_\theta(X; y) \in \mathbb{R}^d$ must be the same, and the loss is bounded as

$$\mathbb{E}_{b \in \mathcal{B}}[|(v_b - f_{\theta(\mathcal{A})})_j|]$$

$$= \frac{1}{|\mathcal{B}|} \sum_{b \in \mathcal{B}} |(v_b - f_{\theta(\mathcal{A})})_j|$$

$$= \frac{1}{|\mathcal{B}|} \left( \sum_{b \in \mathcal{B}_j} |(v_b)_j - (f_{\theta(\mathcal{A})})_j| + \sum_{b \in \mathcal{B}_{\bar{j}}} |(v_b)_j - (f_{\theta(\mathcal{A})})_j| \right)$$

$$\geq \frac{1}{|\mathcal{B}|} \left( \sum_{b \in \mathcal{B}_j \cap \mathcal{Q}} |(v_b)_j - (f_{\theta(\mathcal{A})})_j| + \sum_{b \in \mathcal{B}_{\bar{j}} \cap \mathcal{Q}} |(v_b)_j - (f_{\theta(\mathcal{A})})_j| \right)$$

$$\geq \frac{1}{|\mathcal{B}|} \left( (|\mathcal{B}_j| - |\mathcal{B} \backslash \mathcal{Q}|)|1 - (f_{\theta(\mathcal{A})})_j| + (|\mathcal{B}_{\bar{j}}| - |\mathcal{B} \backslash \mathcal{Q}|)|0 - (f_{\theta(\mathcal{A})})_j| \right)$$

$$\geq \frac{1}{|\mathcal{B}|} \min\{|\mathcal{B}_j| - |\mathcal{B} \backslash \mathcal{Q}|, |\mathcal{B}_{\bar{j}}| - |\mathcal{B} \backslash \mathcal{Q}|\}(|1 - (f_{\theta(\mathcal{A})})_j| + |(f_{\theta(\mathcal{A})})_j|)$$

$$\geq \frac{1}{|\mathcal{B}|} (\min\{|\mathcal{B}_j|, |\mathcal{B}_{\bar{j}}|\} - |\mathcal{B} \backslash \mathcal{Q}|)$$

$$\geq \min\{k/d, 1 - k/d\} - e^{-\Theta(d)}.$$

where the 1st step is by the definition of expectation, the 2nd step is by separating the terms, the 3rd step is by restricting to $\mathcal{Q}$, the 4th and 5th step is by simple algebra, the 6th step is by $|a|+|b| \geq |a+b|$ for $a, b \in \mathbb{R}$, and the last step is by Eq. (11) and Eq. (10). $\qquad \square$

## H  EXTENSION TO NOISY BOOLEAN PROBLEMS

Recall that in Definition 3.3, we define the classical boolean problems. Here we provide a noisy version as an extension.

**Definition H.1** (Learning $k$-bit $p$-Noisy Boolean Functions). *Let $d \geq k \geq 2$ be integers such that $k = \Theta(d)$ and let $\mathcal{B} = \binom{[d]}{k}$ denote the set of all size $k$ subsets of $[d] := \{1, \cdots, d\}$ equipped with the uniform distribution. Let $p \in [0, 1/3]$. Let the $k$ bits in set $b \subseteq [d]$ be $j_1, \ldots, j_k$, and set $t = k/2$. Our goal is to study the noisy $k$-boolean problem for $d$-bit inputs $x = (x_j)_{j=1}^d \sim \mathrm{Unif}(\{0,1\}^d)$, where the target*

$$y_{\mathsf{and}}(x) := \prod_{m=d+1}^{d+t} x'_m, \quad or \quad y_{\mathsf{or}}(x) := 1 - \prod_{m=d+1}^{d+t} (1 - x'_m), \quad with \quad |b| = k,$$

*is determined by the boolean value of the unknown subset of bits $b \in \mathcal{B}$.*

*We can define $k$-bit $p$-Noisy AND function. We suppose that the intermediate bits are noisy, and for each $i \in [t]$ and $l \in [n]$, the vector $x_{d+i} \in \mathbb{R}^n$ is defined by*

$$(x'_{d+i})_l := \begin{cases} (x_{j_{2i-1}})_l (x_{j_{2i}})_l, & \text{with prob. } 1 - p; \quad \text{(correct case)} \\ 1 - (x_{j_{2i-1}})_l (x_{j_{2i}})_l, & \text{with prob. } p. \end{cases}$$

*Similarly, for $k$-bit $p$-Noisy OR function, we have*

$$(x'_{d+i})_l := \begin{cases} 1 - (x_{j_{2i-1}})_l (x_{j_{2i}})_l, & \text{with prob. } 1 - p; \quad \text{(correct case)} \\ (x_{j_{2i-1}})_l (x_{j_{2i}})_l, & \text{with prob. } p. \end{cases}$$

*Given $n$ samples $(x^i, y^i)_{i \in [n]}$, our goal is to predict the size $k$ subset $b \in \mathcal{B}$ deciding the boolean function. In this paper, we denote $x^i \in \mathbb{R}^d$ to be the $i$-th input vector. We denote $x_j \in \mathbb{R}^n$ as $(x_j)_i := (x^i)_j$, i.e. $x_j$ is an $n$-dimensional vector containing the $j$-th bits of all $x^i$, and $y \in \mathbb{R}^n$ as $y_i := \prod_{j \in b} x_j^i$.*

The following theorem can be viewed as a general version of Theorem 4.1. Essentially, Theorem 4.1 only solves the case when $p = 0$.

**Theorem H.2** (Upper Bound: Softmax Attention Provably Solve Definition H.1 with Teacher Forcing). *Let $\epsilon > 0$, and suppose $d$ is a sufficiently large positive integer. Let $k = \Theta(d)$ be an even integer, and set $t = k/2$. Define $\mathcal{B} := \binom{[d]}{k}$ to be the collection of all size-$k$ subsets of $[d]$. Let*

$X := (x_1 \; \cdots \; x_d) \in \mathbb{R}^{n \times d}$ and $E := (x_{d+1} \; \cdots \; x_{d+t}) \in \mathbb{R}^{n \times t}$. Assume $n = \Omega(d^\epsilon)$ and consider any $O(d^{-1-\epsilon/4})$-approximate gradient oracle $\widetilde{\nabla}$. Let the weights be initialized as $W^{(0)} = \mathbf{0}_{d \times t}$. Let $v_b \in \{0,1\}^d$ denote the indicator vector that encodes the Boolean target associated with subset $b \subseteq [d]$. Since ground-truth vector $v_b \in \{0,1\}^d$ is unknown, we define the surrogate function

$$L(W) := \frac{1}{2n} \|\mathrm{Att}_W(X) - E\|_F^2,$$

instead of the loss $\|2 \cdot \mathrm{Softmax}(W^{(1)})\mathbf{1}_t - v_b\|_\infty$ to find the target weight matrix $W$. Set the learning rate $\eta = \Theta(d^{1+\epsilon/8})$, and choose $\kappa \in [d^{-1}, 1]$ (we set $\kappa = O(d^{-\epsilon/4})$). Let $W^{(1)} := W^{(0)} - \eta \cdot \nabla L(W^{(0)})$ be the one-step gradient update.

Then for any target subset $b \in \mathcal{B}$, the algorithm solves the noisy $k$-Boolean problem (Definition H.1) over $d$-bit inputs. Denote $\phi : \mathbb{R} \to \mathbb{R}$ as

$$\phi(x) := \begin{cases} 0, & \text{if } x \le 0.5d^{\epsilon/8}; \\ 1, & \text{otherwise.} \end{cases}$$

With probability at least $1 - \exp(-\Theta(d^{\epsilon/2}))$ over the randomness in sampling and the affect of noise, the one-step update $W^{(1)} \in \mathbb{R}^{d \times t}$ satisfies:

$$\phi(W^{(1)})\mathbf{1}_t = v_b.$$

*Proof.* Denote $\widehat{z} := \frac{1}{d}\sum_{j=1}^d x_j$ as in Theorem 4.1. The surrogate loss function computes the squared error over the intermediate states $x_{d+1}, \cdots, x_{d+t}$:

$$L(W) := \frac{1}{2n} \sum_{m=d+1}^{d+t} \|\widehat{z} - x_m\|^2.$$

For any $i \in [t]$, we firstly bound the number of indices $l \in [n]$ such that

$$(x'_{d+i})_l = 1 - (x_{j_{2i-1}})_l (x_{j_{2i}})_l.$$

Denote $r_i$ to be the number of indices $l$ satisfying the conditions.

Note that

$$\mu = \mathbb{E}[r_i] = pn$$

Using Chernoff bound (Lemma D.3), we have

$$\Pr[r_i \ge (1+\delta)\mu] \le \exp(-\delta^2 \mu / 3)$$

Choosing $\delta = 0.5$, we have

$$\Pr[r_i \ge 1.5pn] \le \exp(-pn/12) = \exp(-\Theta(d^\epsilon)) \tag{12}$$

As in Theorem 4.1, the gradient of $L$ with respect to each element $w_{j,m}$ at initialization can be computed as

$$\begin{aligned}
\frac{\partial L}{\partial w_{j,m}}(W) &= \frac{1}{n}(\widehat{z}_m - x'_m)^\top \frac{\partial \widehat{z}_m}{\partial w_{j,m}} \\
&= \frac{\sigma_j(w_m)}{n} \langle \widehat{z} - x'_m, x_j - \widehat{z} \rangle \\
&= \frac{1}{nd}(-\langle x'_m, x_j \rangle + \langle x'_m, \widehat{z} \rangle + \langle \widehat{z}, x_j \rangle - \langle \widehat{z}, \widehat{z} \rangle) \\
&:= \frac{1}{nd}(B_1 + B_2 + B_3 + B_4), \tag{13}
\end{aligned}$$

where $\widehat{z}$ is defined as $\widehat{z} = \frac{1}{d}\sum_{j=1}^d x_j$.

Using Eq. (12), we have that with probability at least $1 - \exp(\Theta(d^\epsilon))$, $\delta_1 := |A_1 - B_1| \leq 1.5pn$, $\delta_2 := |A_2 - B_2| \leq 1.5pn$. Let $\delta := \delta_1 + \delta_2$. We also have $B_3 = A_3$ and $B_4 = A_4$.

Combining the computation of $A_1$-$A_4$ in Theorem 4.1, $\frac{\partial L}{\partial w_{j,m}}(W)$ is bounded as

$$\frac{1}{nd}(\sum_{i=1}^4 A_i - \delta) \leq \frac{1}{nd}\sum_{i=1}^4 B_i \leq \frac{1}{nd}(\sum_{i=1}^4 A_i + \delta),$$

which deduce to

$$-\frac{1}{8d}\mathbb{1}_{\mathsf{p}[j]=m} + O(d^{-1-\epsilon/4}) - \frac{3p}{d} \leq \frac{\partial L}{\partial w_{j,m}}(W) \leq -\frac{1}{8d}\mathbb{1}_{\mathsf{p}[j]=m} + O(d^{-1-\epsilon/4}) + \frac{3p}{d}.$$

To guarantee that $1/(8d)$ is dominating the term $3p/d$, we need to make that $3p/d \leq 1/(9d)$. This means, $p \leq 1/3$.

Thus, we have

$$-\frac{1}{72d}\mathbb{1}_{\mathsf{p}[j]=m} + O(d^{-1-\epsilon/4}) \leq \frac{\partial L}{\partial w_{j,m}}(W) \leq -\frac{1}{72d}\mathbb{1}_{\mathsf{p}[j]=m} + O(d^{-1-\epsilon/4}).$$

**Property of Softmax Calculations.** Taking $\eta = \Theta(d^{1+\epsilon/8})$, the updated weights

$$W^{(1)} = \underbrace{W^{(0)}}_{d \times \frac{k}{2}} - \eta \underbrace{\widetilde{\nabla} L(W^{(0)})}_{d \times \frac{k}{2}},$$

become

$$w_{j,m}^{(1)} = d^{\epsilon/8}\mathbb{1}_{\{\mathsf{p}[j]=m\}} + O(d^{-\epsilon/8}).$$

Recall $\phi : \mathbb{R} \to \mathbb{R}$ is denoted as

$$\phi(x) := \begin{cases} 0, & x \leq 0.5d^{\epsilon/8}; \\ 1, & \text{otherwise.} \end{cases}$$

Therefore

$$\phi(w_{j,m}^{(1)}) = \begin{cases} 0, & \mathsf{p}[j] \neq m; \\ 1, & \mathsf{p}[j] = m. \end{cases}$$

Since for each $j \in b$, there's exactly one $m \in [d + t]\backslash[d]$ such that $\mathsf{p}[j] = m$, we deduce that $\phi(W^{(1)})\mathbf{1}_t = v_b$.

This completes the proof. $\qquad\qquad\qquad\qquad\qquad\qquad\qquad\qquad\qquad\qquad\qquad\square$

# I  THE MAJORITY PROBLEM

In this section, we extend our techniques to study the $k$-Majority problem, akin to (Chen et al., 2025) (Note that prior work only studies the hardness result). We also want to remark that, the majority problem we study is more or less a local majority problem (where you take two variables as inputs). Such majority problem is not equivalent to the general majority problem, where the inputs can be arbitrary number of variables. In order to define $k$-Majority problem, we need to firstly define majority function.

**Definition I.1** (The Majority Function). *Let $d \in \mathbb{N}_+$. For $x \in \{\pm 1, 0\}^d$ and $S \subseteq [d]$, the majority function $\mathsf{MAJ} : \{\pm 1, 0\}^d \times 2^{[d]}$ is defined as follows:*

$$\mathsf{MAJ}(x, S) := \begin{cases} +1, & \sum_{j \in S} x_j > 0; \\ 0, & \sum_{j \in S} x_j = 0; \\ -1, & \sum_{j \in S} x_j < 0. \end{cases}$$

*In particular, $\mathsf{MAJ}(x, S)$ is also denoted as $\mathsf{MAJ}(x)$ if $S = [d]$.*

*We define $\mathsf{MAJ2}(x + y) := (x + y)/2$.*

Now, we're ready to define the $k$-Majority problem.

**Definition I.2** (The $k$-Majority Problem). *Suppose $d \geq k \geq 2$ are positive integers. Denote $\mathcal{S}$ to be the set of all $S \subseteq [d]$ with $|S| = k$. Let $S \in \mathcal{S}$ be a fixed subset of $[d]$, but unknown. The $k$-majority problem is to find out the subset $S$ with $n$ $d$-bit inputs:*

$$x := (x_j)_{j=1}^d \sim \mathrm{Unif}(\{\pm 1\}^d) \in \mathbb{R}^d,$$

*and the output $y := \mathsf{MAJ}(x, S) \in \{\pm 1, 0\}$.*

**Teacher Forcing.** Suppose $k$ is an even integer and let $t = k/2$. Let the $k$ bits in set $S \subseteq [d]$ be $j_1, \cdots, j_k$. Let $x' \in \{\pm 1, 0\}^t$ such that $x'_m = \mathsf{MAJ2}(x_{j_{2m-1}}, x_{j_{2m}})$ for $m \in [t]$. The majority function $y = \mathsf{MAJ}(x, S)$ is also computed as

$$y = \mathsf{MAJ}(x').$$

The surrogate loss function computes the squared error over the intermediate states $x'$:

$$L(W) := \frac{1}{2n} \sum_{m=1}^t \|\widehat{z}_m - x'_m\|^2,$$

where $\widehat{z}_m = \sum_{j=1}^d \sigma_j(w_m) x_j$.

**Theorem I.3** (Softmax Attention Provably Solve Definition I.2 with Teacher Forcing). *Let $\epsilon > 0$, and $d > 0$ be a sufficiently large integer. Suppose $k = \Theta(d)$ is an even integer, and let $t = k/2$. Define $\mathcal{S} := \binom{[d]}{k}$ as the collection of $[d]$'s all size-$k$ subsets. Denote the $i$-th input as $x^i$ for $i \in [n]$, and let $x_j \in \mathbb{R}^n$ denote all the $j$-th entries of $x^i$, i.e. $(x_j)_i = (x^i)_j$ for all $i \in [n]$ and $j \in [d]$. Set initialization $W^{(0)} = \mathbf{0}_{d \times t}$, and let $E := (x'_1 \cdots x'_t) \in \mathbb{R}^{n \times t}$. For any target subset $S \in [d]$, the algorithm solves the $k$-majority problem (Definition I.2) over $d$-bit inputs. With probability at least $1 - \exp(-\Theta(d^{\epsilon/2}))$ over the randomness in sampling, the one-step update $W^{(1)} \in \mathbb{R}^{d \times t}$ satisfies:*

$$x^\top \mathsf{nint}(2W^{(1)}) \mathbf{1}_t - \mathsf{MAJ}(x, S) = 0,$$

*for any input $x \in \{\pm 1\}^d$.*

*Proof.* Similar to Theorem 4.1, we compute

$$\frac{\partial L}{\partial w_{j,m}}(W) = \frac{1}{n}(\widehat{z}_m - x'_m)^\top \frac{\partial \widehat{z}_m}{\partial w_{j,m}}$$

$$= \frac{\sigma_j(w_m)}{n} \langle \widehat{z}_m - x'_m, x_j - \widehat{z} \rangle$$

$$= \frac{1}{nd}(-\langle x'_m, x_j \rangle + \langle x'_m, \widehat{z} \rangle + \langle \widehat{z}, x_j \rangle - \langle \widehat{z}, \widehat{z} \rangle)$$

$$:= \frac{1}{nd}(C_1 + C_2 + C_3 + C_4),$$

where $\widehat{z}$ is defined as $\widehat{z} := \frac{1}{d} \sum_{j=1}^d x_j$.

**Analyzing the Interaction Terms.** When $p[j] \neq m$,

$$\langle x_j, x'_m \rangle = \frac{1}{2} \langle x_j, x_{\mathsf{c}_1[m]} + x_{\mathsf{c}_2[m]} \rangle.$$

Then by Lemma E.3, we deduce that with probability at least $1 - p$,

$$|\langle x_j, x'_m \rangle| \leq \kappa,$$

for all $j, m$ such that $\mathsf{p}[j] \neq m$.

Combining the above, we have

$$\frac{\partial L}{\partial w_{j,m}}(W) = \frac{1}{2d} \mathbb{1}_{\mathsf{p}[j]=m} + O(d^{-1}\kappa).$$

**Properties of Softmax Calculations.** Taking $\eta = \Theta(d^{1+\epsilon/8})$, the updated weights $W^{(1)} = \underbrace{W^{(0)}}_{d\times t} - \eta \underbrace{\widetilde{\nabla} L(W^{(0)})}_{d\times t}$ become

$$w_{j,m}^{(1)} = d^{\epsilon/8}\mathbb{1}_{\{\mathsf{p}[j]=m\}} + O(d^{-\epsilon/8}). \tag{14}$$

In particular, for each $j \neq \mathsf{c}_1[m], \mathsf{c}_2[m]$, we have

$$\sigma_j(w_m^{(1)}) = e^{w_{j,m}^{(1)}}/\sum_\alpha e^{w_{\alpha,m}^{(1)}}$$

$$\leq e^{w_{j,m}^{(1)} - w_{\mathsf{c}_1[m],m}^{(1)}}$$

$$\leq \exp(-\Omega(d)),$$

where the 1st step is by definition of softmax function, the 2nd step is by simple algebra, and the 3rd step is by Eq. (14).

Using the property $\sum_{j=1}^d \sigma_j(w_m) = 1$, we can show

$$\sigma_{\mathsf{c}_1[m]}(w_m^{(1)}) + \sigma_{\mathsf{c}_2[m]}(w_m^{(1)}) \geq 1 - \exp(-\Omega(d)).$$

Furthermore,

$$\sigma_{\mathsf{c}_1[m]}(w_m^{(1)})/\sigma_{\mathsf{c}_2[m]}(w_m^{(1)}) = e^{w_{\mathsf{c}_1[m],m}^{(1)} - w_{\mathsf{c}_2[m],m}^{(1)}}$$

$$\leq \exp(O(d^{-\epsilon/8}))$$

$$\leq 1 + O(d^{-\epsilon/8}), \tag{15}$$

where the 1st line is by definition, the 2nd line is by Eq. (14), and the 3rd line is by the inequality $e^t \leq 1 + O(t)$ for small $t > 0$.

Then using symmetric property, we have

$$\sigma_{\mathsf{c}_2[m]}(w_m^{(1)})/\sigma_{\mathsf{c}_1[m]}(w_m^{(1)}) \leq 1 + O(d^{-\epsilon/8}). \tag{16}$$

By Eq. (15) and Eq. (16), we have

$$\frac{1}{2} - O(d^{-\epsilon/8}) \leq \sigma_{\mathsf{c}_1[m]}(w_m^{(1)}), \sigma_{\mathsf{c}_2[m]}(w_m^{(1)}) \leq \frac{1}{2} + O(d^{-\epsilon/8}).$$

**Proof of Loss Function.** Define the function $\mathsf{nint}(\cdot) : \mathbb{R} \to \mathbb{Z} : x \mapsto y$, where $y$ is the closest integer with $x$.

When $x$ is a half integer, we define

$$\mathsf{nint}(x) := x - \frac{1}{2}.$$

Therefore we have

$$\mathsf{nint}(2W^{(1)})_{(j,m)} = \begin{cases} 1, & \mathsf{p}[j] = m; \\ 0, & \text{otherwise}. \end{cases}$$

Then for any input $x \in \mathbb{R}^d$, we have

$$x^\top \mathsf{nint}(2W^{(1)})\mathbf{1}_t - \mathsf{MAJ}(x, S) = 0.$$

This completes the proof. $\qquad\qquad\qquad\qquad\qquad\qquad\qquad\qquad\qquad\qquad\qquad\qquad\square$

