# OpenReview forum: "Minimalist Softmax Attention Provably Learns Constrained Boolean Functions"
_ICLR.cc/2026/Conference — ICLR 2026 Conference Withdrawn Submission_

### Official Review · Reviewer_Fnqp · 2025-10-20

**Soundness:** 2
**Presentation:** 3
**Contribution:** 1
**Rating:** 2
**Confidence:** 3

**Summary:**

Shows upper and lower bounds for the complexity of learning conjunctive or disjunctive boolean expressions with a softmax attention network

**Strengths:**

Paper is clear and understandable, although prone to overstatement in places.

**Weaknesses:**

The results in this paper are a minor variation of Kim and Suzuki (2025), who instantiate the results of Shalev-Schwartz et al (2017) in the realm of transformer networks.   In light of Shalev-Schwartz et al (2017), one expects there are infinitely many architectural variations that support a dichotomy where end-to-end learning fails, but decomposition succeeds.

**Questions:**

How about aiming for a more general result rather reducing the generality of prior work?

---

### Official Review · Reviewer_GdTG · 2025-10-30

**Soundness:** 3
**Presentation:** 3
**Contribution:** 3
**Rating:** 6
**Confidence:** 4

**Summary:**

This paper analyzes the ability of one-layer attention networks to learn a k-sparse AND/OR task, where the input is a sequence of bits, and the output is the AND/OR of a k-subset of those bits. With k linear in the input length and no teacher forcing, an exponential number of samples is required to learn the AND/OR task with this one-head network. On the other hand, with teacher forcing (essentially, only requiring the network to predict AND/OR of pairs rather than the full sequence), a single gradient update with a small sample size suffices with high probability. One interpretation of these results is a supervision gap: attention can't learn k-sparse AND/OR unless the correct subset is identified via supervision.

**Strengths:**

1. Analyzing learning k-sparse AND/OR is well motivated by previous work on learning parity with and without teacher forcing. This is an even simpler setting than that prior work
2. The high level results and structure of the paper are largely clear. The comparison between Theorem 4.1 and 4.3 gives theoretical evidence for a supervision gap on the k-sparse AND/OR problem

**Weaknesses:**

### Interpretation of Results

> In real-world scenarios of this form, relying on end-to-end training alone(with no auxiliary signals) may be doomed to fail

Technically you have not proven that a more complicated model with more heads and layers could not efficiently learn k-AND. Yes, expanding the model class seems like it wouldn't make learning easier, but perhaps there is some kind of complex inductive bias that emerges in larger models (or a lottery ticket effect with multiple heads) that would make learning easier.

While the theoretical contribution of the paper is nice, I think the paper overstates its "practical implications" in 4.3, as most of this discussion is quite vague. Could you be more specific about how you think the findings here could influence the design of intermediate supervision, auxiliary losses, or curricula?
### Theorem 4.1

Is it correct to re-interpret Theorem 4.1 to say you can learn k-AND (without teacher forcing) with k=2? That's essentially what's happening; you're just doing t/2 of them in parallel.

Does this generalize to k=O(1)? The difficulty comes when k = Theta(d), and teacher forcing helps because it reduces Theta(d) to O(1). Can you explain briefly why the Theorem 4.1 proof breaks down without teacher forcing when k is not O(1)? I.e., how can we see that softmax will not concentrate on the right subset of size k in this case?

Theorem 4.1 proof: say something about the interpretation of $p$ and $p[j] = m$ in natural language, which will make the proof a lot clearer.
### Theorem 4.2

Something is weird about the Theorem 4.3 statement: $n = e^{\Theta(d)}$ is a precondition, so technically with polynomial samples (or less), the theorem says nothing.

Beyond this issue, it would be nice to give some sense of the techniques used to prove this, and what changes from the teacher forcing case. Is it a statistical query learning argument? What properties of the transformer/learning algorithm are needed to obtain it? It's hard to appreciate the discussion after the theorem about whether lower bounds are possible in the prior paper's setup without this.

Finally, it seems like we can understand the difference between the two theorems as going from k=2 to k=Theta(d), both without teacher forcing. Is this right, and can you better characterize what's going on with this transition?

**Questions:**

Give citations for the following claims:

> In contrast, under ordinary end-to-end training (only input-label pairs, no intermediate hints) no algorithm running in poly(d) time can recover the same function, even when given eΩ(d) examples

> Prior work has likewise shown that even small Transformers can emulate complex computations by appropriate setting of their weights.

Abstract: definite d as sequence length here

You claim that the value matrix V is set as the identity matrix I (d x d), where d is effectively the sequence length. But the value matrix would map the feature axis (which is missing in your simplified model but could be taken to be k=1), so your claim is not exactly write as stated. Rather, you implicitly have a value weight matrix of size 1 x 1, or a value activation matrix V = X (not V as d x d identity).

Move definition of teacher forcing to Section 3

---

### Official Review · Reviewer_p1rv · 2025-10-30

**Soundness:** 2
**Presentation:** 2
**Contribution:** 2
**Rating:** 2
**Confidence:** 4

**Summary:**

The paper investigates when a single-head, one-layer softmax-attention module can or cannot learn a $k$-bit monotone Boolean function (AND/OR). Two regimes are compared:

1. **Oracle “teacher-forcing” supervision:**
   The loss includes intermediate targets
   $$
   E = {x_{j_{2i-1}}x_{j_{2i}}}_{i=1}^{k/2},
   $$
   explicitly revealing the relevant bits. Under this oracle signal, one gradient update from $W^{(0)}=0$ produces weights $W^{(1)}$ such that
   $$
   |2,\mathrm{Softmax}(W^{(1)})\mathbf{1}*t - v_b|*\infty = O(d^{-\varepsilon/8}),
   $$
   i.e., exact mask recovery after one step (Theorem 4.1 / App. F).

2. **End-to-end (no intermediate targets):**
   Theorem 4.3 claims a lower bound showing that no polynomial-time learner can identify $b$ even with $n = e^{\Theta(d)}$ samples.

The paper frames this as exposing a *supervision gap* between “unsolvable” and “one-step solvable” regimes.

**Strengths:**

1. **One-step gradient derivation under oracle supervision.**
   The paper rigorously derives that, given perfect intermediate supervision, a single gradient step from
   $$
   W^{(0)} = 0 \quad\Rightarrow\quad
   W^{(1)}_{j,m} = \frac{d^{\varepsilon/8}}{8}\mathbf{1}{p[j]=m} + O(d^{-\varepsilon/8}),
   $$
   leads to an attention mask satisfying
   $$
   |2,\mathrm{Softmax}(W^{(1)})\mathbf{1}*t - v_b|*\infty = O(d^{-\varepsilon/8}),
   $$
   meaning the model exactly identifies the relevant subset in one update.
   The algebraic and probabilistic steps (especially the decomposition into $A_1,\dots,A_4$ terms and the use of Hoeffding and Chernoff concentration) are sound and clearly written.

2. **Clear illustration of supervision-driven learnability.**
   The theoretical construction neatly captures how introducing oracle intermediate targets collapses the combinatorial hypothesis space from $\binom{d}{k}$ possible supports to a tractable one-step solution.
   It serves as a clean example of how information shaping affects optimization landscapes.

3. **Strong pedagogical and expository value.**
   Even though the assumptions are unrealistic, the result helps clarify how intermediate supervision alters gradient geometry.
   The explicit connection between supervision, signal alignment, and the scale of gradient entries ($\Omega(d^{\varepsilon/8})$ vs. $O(d^{-1})$) makes it easy for readers to see why teacher-forced learning converges instantly.

4. **Concise theoretical exposition.**
   The proofs in Appendix F are complete, consistent, and don't seem to contain many shortcuts. The authors also formalize the concentration lemmas (Appendix E) instead of invoking them informally (which I love and appreciate).
   The clarity of the one-step analysis makes it a useful reference for future work on alignment and inductive bias in attention-based learners.

5. **Conceptual relevance.**
   The study reinforces an important theoretical theme that expressivity is cheap while learnability is expensive, consistent with PAC and statistical query frameworks but contextualized for the attention mechanism.
   This conceptual link, though not novel, is clearly presented and theoretically valuable.

**Weaknesses:**

1. **Misleading framing of contributions.**
   The abstract and introduction claim a “fundamental computational limit” for monotone AND/OR functions, yet the paper never proves a formal lower bound for these functions. The only lower bound provided is a generic PAC-style hardness lemma of the form
   $
   \mathbb{E}*{b,x}!\left[\min_j |(v_b - f*\theta(x,y))_j|\right]
   \ge \min!\left{\frac{k}{d}, 1 - \frac{k}{d}\right} - e^{-\Theta(d)},
   $

   which is derived directly from parity and majority analyses in Kim & Suzuki (2025) and Chen et al. (2025).
   In Section 4.2, the authors even acknowledge that extending these parity-based frameworks to AND/OR is “unlikely.” Despite this, the abstract still describes the result as a “provably impossible” case for standard training, which is not supported by their proofs.

2. **Incorrect and inconsistent use of terminology.**
   The paper repeatedly uses the term *teacher forcing* to describe a loss that provides oracle-level access to intermediate features
   $$
   E = {x_{j_{2i-1}}x_{j_{2i}}}, \quad p(j_{2i-1}) = p(j_{2i}) = i,
   $$
   which is far stronger than teacher forcing in sequence models, where the supervision involves ground-truth outputs at previous time steps.
   Here, the loss discloses the hidden support $b$ directly, collapsing the learning problem into a trivial optimization. This should be described as *oracle intermediate supervision* rather than teacher forcing.
   Similarly, phrases such as “no supervision” and “unsupervised” are used where the model is still trained on labeled data, which confuses readers about the experimental regime.

3. **Overstatement of novelty relative to prior work.**
   The core idea that intermediate supervision transforms an intractable search into a tractable learning problem has already been developed in PAC and SQ frameworks (Kearns, 1998; Blum et al., 1994) and in the Transformer context by Kim & Suzuki (2025).
   Those works already establish that gradient-based learners fail on parity-like problems without intermediate reasoning steps. This paper rephrases those results for AND/OR functions without introducing a new analytical technique or complexity class. The “one-step” result merely exploits the oracle supervision’s direct alignment with the gradient.

4. **Logical inconsistency between claims and setup.**
   The paper simultaneously asserts that single-head attention *cannot* solve AND/OR functions and that under teacher forcing it *can*.
   Since both claims concern the same architecture, the difference lies entirely in the supervision regime, not in model capability. The term “unsolvable” therefore misrepresents the finding. What is shown is a *learnability gap* under differing supervision signals, not an expressivity limit.

5. **Mathematical inaccuracies and omissions.**
   Several technical sections contain small but significant errors:

   * In Appendix H, the noise constraint is mis-solved: the inequality $3p/d \le 1/(9d)$ implies $p \le 1/27$, not $p \le 1/3$.
   * In Appendix G, the statement “$1 - \exp(\Theta(d))$” should be “$1 - \exp(-\Theta(d))$.”
   * The union bound in Appendix E, $6p^4/d \le p$, holds only if $p = \exp(-d^{\varepsilon/2})$.
   * Notational drift occurs between $z$, $z_b$, and $z_{bm}$, and the index range “$d < m \le 2d - 1$” is incorrect and should read “$m \in (d, d + t]$.”
     These errors do not change the overall trend but detract from mathematical precision.

6. **Lack of calibration between surrogate and true task loss.**
   The loss used for optimization,
   $
   L(W) = \frac{1}{2n}|\mathrm{Att}_W(X) - E|*F^2,
   $

   minimizes the distance between attention outputs and oracle intermediates, not the actual classification objective $\mathrm{AND/OR}(x_b)$.
   The authors never establish a link between convergence in this surrogate loss and generalization on the true Boolean label, nor do they prove a calibration bound such as
   $\mathrm{excess}*{0\text{-}1} \le \Phi(|\mathrm{Softmax}(W)\mathbf{1}*t - v_b|*\infty)$
   for any explicit function $\Phi$.

7. **Misleading comparison to Chain-of-Thought (CoT) supervision.**
   The claim that one gradient step “replaces” multi-step CoT reasoning from Kim & Suzuki (2025) is inaccurate.
   In CoT, intermediate **reasoning tokens** provide linguistic structure; here, the supervision is mathematically equivalent to providing the **ground-truth latent variables**.
   The two regimes differ in kind, not in training length. Equating them inflates the significance of the one-step result.

8. **Limited originality and lack of broader implications.**
   While the one-step result is analytically neat, it relies on unrealistic assumptions that prevent practical relevance.
   No new theoretical mechanism is introduced beyond restating that oracle supervision gives the gradient perfect alignment with the target subset.
   The work neither expands the statistical query hardness class nor presents new upper bounds for realistic training scenarios.

**Questions:**

1. **Clarify what supervision actually occurs under “teacher forcing.”**
   In your setup, the learner is given intermediate products
   $$
   E = {x_{j_{2i-1}}x_{j_{2i}}}*{i=1}^{t}, \qquad p(j*{2i-1}) = p(j_{2i}) = i,
   $$
   which explicitly reveal the hidden subset $b$ and its column mapping $p:[d]\to[t]$.
   This is closer to *oracle intermediate supervision* than to standard teacher forcing used in sequence models, where the model is conditioned on ground-truth outputs $y_{t-1}$ but not on latent features.
   What supervision regime would actually be **analogous** to teacher forcing in your Boolean setting? For instance, could teacher forcing correspond to conditioning on partial evaluations of $\mathrm{AND/OR}(x_b)$ (e.g., partial conjunctions) rather than providing $E$ directly? Please formalize what you consider “teacher forcing” mathematically.

2. **Establish a link between your surrogate loss and the true Boolean objective.**
   You minimize
   $$
   L(W)=\frac{1}{2n}|\mathrm{Att}_W(X)-E|*F^2,
   $$
   where $E$ is the oracle signal, but the actual task is to minimize classification error on $y=\mathrm{AND/OR}(x_b)$.
   Can you derive a calibration inequality showing how convergence in the surrogate loss implies convergence in the Boolean label loss?
   Specifically, can you prove that
   $$
   \mathrm{excess}*{0\text{-}1} \le \Phi!\left(|\mathrm{Softmax}(W)\mathbf{1}*t - v_b|*\infty\right)
   $$
   for an explicit function $\Phi$? If not, please clarify why minimizing $L(W)$ can be interpreted as “learning AND/OR.”

3. **Quantify the gradient informativeness gap between supervised and unsupervised regimes.**
   You claim that, without the oracle signal $E$, the gradient averages over all $\binom{d}{k}$ hypotheses and is “nearly uninformative.”
   Please compute the expected gradient under end-to-end training at initialization:
   $$
   g_{\text{end}} = \mathbb{E}\bigl[\nabla_W L_{\text{end-to-end}}(W^{(0)})\bigr],
   $$
   and compare it to the oracle-supervised gradient
   $$
   g_{\text{oracle}} = \mathbb{E}\bigl[\nabla_W L_{\text{oracle}}(W^{(0)})\bigr].
   $$
   How small is $\langle g_{\text{end}}, v_b \rangle$ relative to $\langle g_{\text{oracle}}, v_b \rangle$ as a function of $d$ and $k$?
   A quantitative bound here would concretely support your “intractability under standard training” claim.

4. **Formally define and justify the lower-bound claim for AND/OR.**
   Theorem 4.3 is adapted from parity/majority frameworks where balanced output probabilities ($\Pr[y=1]=1/2$) enable binomial-coefficient cancellation.
   For monotone AND/OR, this symmetry fails.
   Can you provide an explicit *information-theoretic* or *SQ-dimension* argument that yields a genuine lower bound for AND/OR under $\mathsf{Unif}({0,1}^d)$?
   For example, show via Fano’s inequality that
   $$
   n \ge \frac{\log\binom{d}{k} - \log 2}{I((X,Y);b)},
   $$
   or compute the SQ tolerance $\tau$ for which no polynomial-query learner can recover $b$.
   Without such a derivation, the term “provably impossible” is misleading- please provide or correct the corresponding proof.

---

### Official Review · Reviewer_sgiH · 2025-10-31

**Soundness:** 2
**Presentation:** 3
**Contribution:** 2
**Rating:** 4
**Confidence:** 4

**Summary:**

The paper studies when a single-head softmax attention layer, without FFNs or depth, can learn (k)-bit monotone Boolean functions, AND/OR over an unknown subset ($b\subset[d]$) with ($|b|=k=\Theta(d))$. The model is parametrized by a matrix $W\in\mathbb{R}^{d\times t}$ with content-independent scores (the authors reparametrize $K^\top Q$ as $W$ and set $V=I$), so that $\mathrm{Att}W(X)=X,\mathrm{softmax}(W)\in\mathbb{R}^{n\times t}$ for the design matrix $X\in\mathbb{R}^{n\times d}$ of inputs. Under an idealized teacher-forcing loss that supervises intermediate pairwise products of the hidden relevant bits, known as hints in the NAR community, the paper proves that one gradient step from $W^{(0)}=0$ already concentrates attention on the true support, so each relevant bit receives $\approx 1/2$ mass in exactly one column, and non-relevant bits get exponentially small mass. Thus, aggregating across columns allows recovery of (b) with $\ell_\infty$ error $O(d^{-\varepsilon/8})$ using $n=\Omega(d^\varepsilon)$ samples (Theorem 4.1 / Appendix F). In contrast, under end-to-end training without hints, the paper argues one cannot identify the support efficiently with typical random inputs, all labels are almost always zero for AND/OR, so the data provides negligible information and any learner’s per-coordinate error remains bounded away from zero (Theorem 4.3 / Appendix G). I believe the main message of the paper is that capacity is present even in a minimalist attention module, but intermediate supervision (teacher forcing / CoT-like signals) give us true learnability (as apposed to memorisation).

**Strengths:**

On the originality side, I found the deep analysis on hits (\w vs \wo) for monotone (k)-bit AND/OR, complementing recent CoT body of research. The proof strategy is also smart and I like it. The noisy-hint extension (like a causal intervention) and a local-majority variant broaden the conceptual message beyond the exact noiseless AND/OR idealization. I believe the paper’s thesis and experimentation is clearly articulated and timely. Even with strong idealizations, the result is a minimal testbed for exploring supervision signals.

**Weaknesses:**

1. I am afraid the analysis looks at the issue from a very idealized perspective.  If my understanding is correct, the attention is content-independent, meaning that a free parameter W replaces $K^\top Q$ and does not depend on X! This makes the module a fixed positional mixer, not self-attention and it cannot adapt attention to the input content. Thus, claims like “one-head softmax attention learns high-dimensional Boolean concepts” risk being over-interpreted, meaning that what is learned here is the support across the dataset, not an input-conditional mapping.
2. I believe teacher forcing is stronger than CoT. The hints are explicit pairwise products of the unknown relevant bits, meaning that they encode the support partition $p[j]=m$ directly. The authors acknowledge this as “idealized” and stronger than typical CoT, but many statements (e.g., “universal Boolean learner in one shot”) read too broadly. I think a careful discussion of what information the hints leak (they leak which bits interact) and how far this is from realistic CoT supervision is needed.
3.Since the presented bound is primarily information-theoretic, Ithink computational hardness is mischaracterized. Theorem 4.3 does not condition on any complexity assumption nor analyze optimization landscapes; it shows that under the random-input distribution, with (n) well below $2^{k}$ the observations are almost all zeros, so no algorithm (efficient or not) can identify b well. That is a statistical identifiability barrier, not a computational one. Yet the paper repeatedly frames it as computational intractability and loss-landscape traps, which is not supported by the proof given. I think a proper clarification is needed here.
4. There are also several proof-level inconsistencies (see questions)
5. I think an important and relevant body of research is not being discussed in this paper, namely Neural Algorithmic Reasoning (NAR) that studies when neural networks can execute or emulate algorithms, with and without intermediate supervision (hints/execution traces), and how alignment between the model’s computation and the target algorithm affects sample efficiency and out-of-distribution generalization. I think positioning your results within NAR would increase the soundness of your work and strengthen the narrative. These are some relevant studies:
- Veličković et al. (LoG 2020), Neural execution of graph algorithms.
- Rodionov et al. (NeurIPS 2023), Neural algorithmic reasoning without intermediate supervision.
- Hashemi et al. (NeurIPS 2025), Tropical Attention: Neural Algorithmic Reasoning for Combinatorial Algorithms
- Li et al. (2025), Circuit transformer: A transformer that preserves logical equivalence
- Rodionov et al. (2025). Discrete neural algorithmic reasoning
- Li et al (NeurIPS 2024). Open-book neural algorithmic reasoning.

**Questions:**

0. First, questions posed in weaknesses.
1. After one step, the relevant–irrelevant logit gap is $\Theta(d^{\varepsilon/8})$ (Eq. (5)), so non-relevant softmax mass should be $\exp(-\Theta(d^{\varepsilon/8}))$, but the text repeatedly says $\exp(-\Theta(d))$ (e.g., Eq. (6) and subsequent bounds). I think this needs correction.
2. Lemma E.2 is stated for $r\in[2]$ over {0,1}, but the proof and Theorem F.1 use triple products (an $r=3$ interaction) when $p[j]\neq m$. The union bound in E.2 even counts $|I_3|$, suggesting the intent was $r\le 3$. Can the authors clarify?
3. In Appendix G, I think the bound $\Pr[y=0^n]\ge 1-n2^{-k}=1-\exp(\Theta(d))$ misses a minus in the exponent. I think it needs correction. I saw some more of these missing. A complete double checking is required I believe. For example, $ẑ$, $ẑ_m$, $x_m$ (where $m$ indexes the hint columns) are used interchangeably across sections, and indices shift between $[d]$ and $[d+t]$. Maybe, an index table would help.

---

### Official Review · Reviewer_hU2w · 2025-11-01

**Soundness:** 4
**Presentation:** 3
**Contribution:** 3
**Rating:** 8
**Confidence:** 3

**Summary:**

The work studies the computational limits of the softmax-attention mechanism, studying its ability to learn a k-bit Boolean AND/OR function where only a subset of the given bits are relevant. They show that such Boolean functions are hard to learn in general with just a single-head softmax-attention mechanism. However, with *teacher forcing* the learning process can be informed of the relevant input bits, such that learning a k-bit Boolean function becomes possible with a single gradient step. This indicates the importance of not only considering the architecture's expressivity, but also the details of the learning process. In summary, as a contribution, this work improves known bounds, expanding our understanding of the attention architecture.

**Strengths:**

The majority of the work is presented very clearly. For example, the research question is made very explicit: "can gradient descent training on input-output examples learn to attend to the correct k bita and reliably compute the AND/OR?". Overall, as an outsider to this domain, I found the paper very insightful and qualitative. Only the soundness of the contribution I could not verify.

The work is well motivated, expanding our knowledge of what a softmax-attention mechanism can and cannot do, stressing that while the architecture's expressivity is important, so is the learning process.

**Weaknesses:**

Novelty. While certainly different, the work heavily connects to [Kim & Suzuki, 2025] who have studied the ability of the transformer architecture to learn the parity function. This work instead studies an easier Boolean problem, expanding insights of the architecture. The work certainly appears novel, but I was unable to assess this with high confidence given it is outside my expertise.

The presentation of the contributions was very clear. However, minor remark, some parts of the contribution are repeated multiple times to a degree that reads very repetitively.

**Questions:**

It took a while to understand what "teacher forcing" means, as it is used frequently before explaining what it means.

Typo: "and For" (line 147)

---

### Note · Authors · 2025-12-24

**Comment:**

We have decided to withdraw our paper from ICLR 2026. For 2 main reasons:

1. Several reviews are very helpful and point to substantial revisions we plan to make (sharper framing/positioning, tighter statements/proofs...etc). These revisions require some time to do properly.

2. The recent OpenReview incident disrupted the rebuttal period. By the time we were ready to post a revision and rebuttal, the active rebuttal window had passed, so a meaningful rebuttal is no longer possible in this cycle.

Given this, we believe it is simpler and fairer for everyone if we withdraw, incorporate the feedback, and aim for a future submission instead.

We thank the reviewers for all your constructive comments! We also thank the program chairs for their time and effort during this difficult period.

**Withdrawal Confirmation:**

I have read and agree with the venue's withdrawal policy on behalf of myself and my co-authors.